# Variational Rectified Flow Matching

Pengsheng Guo [1]   Alexander G. Schwing

## Abstract

We study Variational Rectified Flow Matching, a framework that enhances classic rectified flow matching by modeling multi-modal velocity vector-fields. At inference time, classic rectified flow matching 'moves' samples from a source distribution to the target distribution by solving an ordinary differential equation via integration along a velocity vector-field. At training time, the velocity vector-field is learnt by linearly interpolating between coupled samples one drawn from the source and one drawn from the target distribution randomly. This leads to "ground-truth" velocity vector-fields that point in different directions at the same location, i.e., the velocity vector-fields are multi-modal/ambiguous. However, since training uses a standard mean-squared-error loss, the learnt velocity vector-field averages "ground-truth" directions and isn't multi-modal. In contrast, variational rectified flow matching learns and samples from multi-modal flow directions. We show on synthetic data, MNIST, CIFAR-10, and ImageNet that variational rectified flow matching leads to compelling results.

## 1. Introduction

Diffusion models (Ho et al., 2020; Song et al., 2021a;b) and flow matching (Liu et al., 2023; Lipman et al., 2023; Albergo & Vanden-Eijnden, 2023; Albergo et al., 2023) have been remarkably successful in recent years. These techniques have been applied across domains from computer vision (Ho et al., 2020) and robotics (Kapelyukh et al., 2023) to computational biology (Guo et al., 2024) and medical imaging (Song et al., 2022).

Flow matching (Lipman et al., 2023; Liu et al., 2023; Albergo & Vanden-Eijnden, 2023) can be viewed as a continuous time generalization of classic diffusion models (Albergo

et al., 2023; Ma et al., 2024). Those in turn can be viewed as a variant of a hierarchical variational auto-encoder (Luo, 2022). At inference time, flow matching 'moves' a sample from a source distribution to the target distribution by solving an ordinary differential equation via integration along a velocity vector-field. To learn this velocity vector-field, at training time, flow matching regresses to a constructed vector-field/flow connecting any sample from the source distribution — think of the data-domain positioned at time zero — to any sample from the target distribution attained at time one. Notably, in a 'rectified flow,' the samples from the source and target distribution are connected via a straight line as shown in Figure 1(a). Inevitably, this leads to multi-modality/ambiguity, i.e., flows pointing in different directions at the same location in the data-time space, as illustrated for a one-dimensional data-domain in Figure 1(a). Since classic rectified flow matching employs a standard squared-norm loss to compare the predicted velocity vector-field to the constructed velocity vector-field, it does not capture this multi-modality. Hence, rectified flow matching aims to match the source and target distribution in alternative ways. This is illustrated in Figure 1(b).

To enable rectified flow matching to capture this multi-modality in the data-time space, we study *variational rectified flow matching*. Intuitively, variational rectified flow matching introduces a latent variable that permits to disentangle multi-modal/ambiguous flow directions at each location in the data-time space. This approach follows the classic variational inference paradigm underlying expectation maximization or variational auto-encoders. Indeed, as shown in Figure 1(c), variational rectified flow matching permits to model flow trajectories that intersect. This leads to learned trajectories that more closely align with the "ground-truth" flow. The latent variable can also be used to disentangle different directions.

Note that flow matching, diffusion models, and variational auto-encoders are all able to capture multi-modality in the data-domain, as one expects from a generative model. Importantly, variational rectified flow matching differs in that *it also models multi-modality in the data-time space*. This enables different flow directions at the same data-time space point, allowing the resulting flows to intersect at that location.

[1]Apple. Correspondence to: Pengsheng Guo <pengsheng_guo@apple.com>.

*Proceedings of the 42nd International Conference on Machine Learning*, Vancouver, Canada. PMLR 267, 2025. 

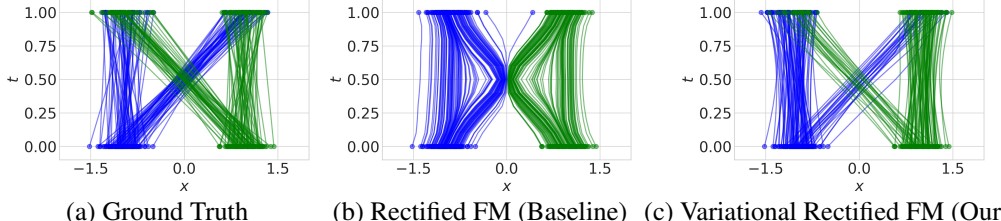

(a) Ground Truth      (b) Rectified FM (Baseline)    (c) Variational Rectified FM (Ours)

Figure 1: Intuition and motivation: Rectified flow matching randomly couples source data and target data samples, as illustrated in panel (a). This leads to velocity vector-fields with ambiguous directions. Panel (b) shows that the classic rectified flow matching averages ambiguous targets, which leads to curved flows. In contrast, the proposed variational rectified flow matching is able to successfully model ambiguity which leads to less curved flows as depicted in panel (c).

We demonstrate the benefits of variational rectified flow matching across various datasets and model architectures. On synthetic data, our method more accurately models data distributions and better captures velocity ambiguity. On MNIST, it enables controllable image generation with improved quality. On CIFAR-10, our approach outperforms classic rectified flow matching across different integration steps. Lastly, on ImageNet, our method consistently improves the FID score of SiT-XL (Ma et al., 2024).

In summary, our contribution is as follows: we study the properties of variational rectified flow matching, and, along the way, offer an alternative way to interpret the flow matching procedure.

## 2. Preliminaries

Given a dataset $\mathcal{D} = \{(x_1)\}$ consisting of data samples $x_1$, e.g., an image, generative models learn a distribution $p(x_1)$, often by maximizing the likelihood. In the following we discuss how this distribution is learnt with variational auto-encoders and rectified flow matching, and why the corresponding modeled data distribution is multi-modal.

### 2.1. Variational Auto-Encoders (VAEs)

Variational inference generally and variational auto-encoders (VAEs) (Kingma & Welling, 2014) specifically have been shown to learn multi-modal distributions. This is achieved by introducing a latent variable $z$. At inference time, a latent $z$ is obtained by sampling from the prior distribution $p(z)$, typically a zero mean unit covariance Gaussian. A decoder which characterizes a distribution $p(x_1|z)$ over the output space is then used to obtain an output space sample $x_1$.

At training time, variational auto-encoders use an encoder to compute an approximate posterior distribution $q_\phi(z|x_1)$ over the latent space. As the approximate posterior distribution is only needed at training time, the data $x_1$ can be leveraged. Note, for the approximate posterior distribution a Gaussian with parameterized mean and covariance is often used. A sample from this approximate posterior distribution

is then used as input in the distribution $p_\theta(x_1|z)$ characterized by the decoder. The loss encourages a high probability of the output space samples while favoring an approximate posterior distribution $q_\phi(z|x_1, c)$ that is similar to the prior distribution $p(z)$. To achieve this, formally, VAEs maximize a lower-bound on the log-likelihood, i.e.,

$$\mathbb{E}_{x_1 \sim \mathcal{D}} \log p(x_1)$$
$$\geq \mathbb{E}_{x_1 \sim \mathcal{D}} \left[ \mathbb{E}_{z \sim q_\phi} \left[ \log p_\theta(x_1|z) \right] - D_{\mathrm{KL}}(q_\phi(\cdot|x_1)|p(\cdot)) \right].$$

### 2.2. Rectified Flow Matching

For flow matching, at inference time, a source distribution $p_0(x_0)$ is queried to obtain a sample $x_0$. This is akin to sampling of a latent variable from the prior in VAEs. Different from VAEs which perform a single forward pass through the decoder, in flow matching, the source distribution sample $x_0$ is used as the boundary condition for an ordinary differential equation (ODE). This ODE is 'solved' by pushing the sample $x_0$ forward from time zero to time one via integration along a trajectory specified via a learned velocity vector-field $v_\theta(x_t, t)$ defined at time $t$ and location $x_t$, and commonly parameterized by deep net weights $\theta$. Note, the velocity vector-field is queried many times during integration. The likelihood of a data point $x_1$ can be assessed via the instantaneous change of variables formula (Chen et al., 2018; Song et al., 2021b; Lipman et al., 2023),

$$\log p_1(x_1) = \log p_0(x_0) + \int_1^0 \operatorname{div} v_\theta(x_t, t) dt, \quad (1)$$

which is commonly (Grathwohl et al., 2018) approximated via the Skilling-Hutchinson trace estimator (Skilling, 1989; Hutchinson, 1990). Here, $\operatorname{div}$ denotes the divergence vector operator.

Intuitively, by pushing forward samples $x_0$, randomly drawn from the source distribution $p_0(x_0)$, ambiguity in the data domain is captured as one expects from a generative model.

At training time the parametric velocity vector-field $v_\theta(x_t, t)$ needs to be learnt. For this, coupled sample pairs $(x_0, x_1)$ are constructed by randomly drawing from the source and the target distribution, often independently from each other.

A coupled sample $(x_0, x_1)$ and a time $t \in [0, 1]$ is then used to compute a time-dependent location $x_t$ at time $t$ via a function $\phi(x_0, x_1, t) = x_t$. Recall, rectified flow matching uses $x_t = \phi(x_0, x_1, t) = (1 - t)x_0 + tx_1$. Interpreting $x_t$ as a location, intuitively, the "ground-truth" velocity vector-field $v(x_0, x_1, t)$ is readily available via $v(x_0, x_1, t) = \partial\phi(x_0, x_1, t)/\partial t$, and can be used as the target to learn the parametric velocity vector-field $v_\theta(x_t, t)$. Concretely, flow matching learns the parametric velocity vector field $v_\theta(x_t, t)$ by matching the target via an $\ell_2$ loss, i.e., by minimizing w.r.t. trainable parameters $\theta$ the objective

$$\mathbb{E}_{t,x_0,x_1}\left[\|v_\theta(x_t, t) - v(x_0, x_1, t)\|_2^2\right].$$

Consider two different couplings that lead to different "ground-truth" velocity vectors at the same data-time space point $(x_t, t)$. The parametric velocity vector-field $v_\theta(x_t, t)$ is then asked to match/regress to a different target given the same input $(x_t, t)$. This leads to averaging and the optimal functional velocity vector-field $v^*(x_t, t) = \mathbb{E}_{\{(x_0,x_1,t):\phi(x_0,x_1,t)=x_t\}}[v(x_0, x_1, t)]$. Hence, multi-modality in the data-time space is not captured. In the following we discuss and study a method that is able to model this multi-modality.

## 3. Variational Rectified Flow Matching

Our goal is to capture the multi-modality inherent in "ground-truth" velocity vector-fields obtained from typically used couplings $(x_0, x_1)$ that connect source distribution samples $x_0 \sim p_0$ with target data samples $x_1 \in \mathcal{D}$. Here, $p_0$ is a known source distribution and $\mathcal{D}$ is a considered dataset. This differs from classic rectified flow matching which does not capture this multi-modality even for simple distributions as shown in Figure 1 and as discussed in Section 2. The struggle to capture multi-modality leads to velocity vector fields that may be more curve and consequently more difficult to integrate at inference time. In turn, this leads to distributions that may not fit the data as well. We will show evidence for both, more difficult integration and less accurately captured data distributions in Section 4.

To achieve our goal we combine rectified flow matching and variational auto-encoders. In the following we first discuss the objective before detailing training and inference.

### 3.1. Objective

The goal of flow matching is to learn a velocity vector-field $v_\theta(x_t, t)$ that transports samples from a known source distribution $p_0$ at time $t = 0$ to samples from a commonly unknown probability density function $p_1(x_1)$ at time $t = 1$. The probability densities $p_0, p_1$ and the velocity vector-field $v_\theta$ are related to each other via the transport problem

$$\frac{\partial \log p_t(x_t)}{\partial t} = -\operatorname{div} v_\theta(x_t, t), \quad (2)$$

or its integral form given in Equation (1).

Solving the partial differential equation given in Equation (2) in general analytically is challenging, even when assuming availability of the probability density functions, i.e., when addressing a classic boundary value problem.

However, if we assume the probability density functions to be Gaussians and if we restrict the velocity vector-field to be constant, i.e., of the simple parametric form $v_\theta(x_t, t) = \theta$, we can obtain an analytic solution. This is expressed in the following claim:

**Claim 1.** *Consider two Gaussian probability density functions* $\tilde{p}_0 = \mathcal{N}(\xi_0; x_0, I)$ *and* $\tilde{p}_1 = \mathcal{N}(\xi_1; x_1, I)$ *with mean* $x_0$ *and* $x_1$ *respectively. Assume a constant velocity vector-field* $v_\theta(\xi_t, t) = \theta$. *Then* $\theta = x_1 - x_0$ *solves the partial differential equation given in Equation* (2) *and its integral form given in Equation* (1) *and* $x_t = (1 - t)x_0 + tx_1$.

**Proof:** Given the constant velocity vector-field $v_\theta(\xi_t, t) = \theta$, we have $\int_1^0 \operatorname{div} v_\theta(\xi_t, t)dt \equiv 0$. Plugging this and both probability density functions into Equation (1) yields $(\xi_0 - x_0)^2 - (\xi_1 - x_1)^2 \equiv 0 \,\forall \xi_0, \xi_1$. Using $\xi_1 = \xi_0 + \int_0^1 v_\theta(\xi_t, t)dt = \xi_0 + \theta$ leads to $(\xi_0 - x_0)^2 - (\xi_0 - x_1 + \theta)^2 \equiv 0 \,\forall \xi_0$ which is equivalent to $(x_1 - x_0 - \theta)(2\xi_0 - x_0 - x_1 + \theta) \equiv 0 \,\forall \xi_0$. This can only be satisfied $\forall \xi_0$ if $\theta = x_1 - x_0$, leading to $x_t = x_0 + t\theta = (1 - t)x_0 + tx_1$, which proves the claim. ∎

The arguably very simple setup in Claim 1 provides intuition for the objective of classic rectified flow matching and offers an alternative way to interpret the flow matching procedure. Specifically, instead of two Gaussian probability density functions $\tilde{p}_0$ and $\tilde{p}_1$, we assume the real probability density functions for the source and target data are composed of Gaussians centered at given data points $x_0$ and $x_1$ respectively, e.g., $p_0(\xi_0) = \sum_{x_0 \in \mathcal{S}} \mathcal{N}(\xi_0; x_0, I)/|\mathcal{S}|$. Moreover, importantly, let us assume that the velocity vector-field $v_\theta(x_t, t)$ at a data-time space location $(x_t, t)$ is characterized by a uni-modal standard Gaussian

$$p(v|x_t, t) = \mathcal{N}(v; v_\theta(x_t, t), I)$$

with a parametric mean $v_\theta(x_t, t)$. Maximizing the log-likelihood of the empirical "velocity data" is equivalent to the following objective

$$\begin{aligned} &\mathbb{E}_{t,x_0,x_1}\left[\log p(x_1 - x_0|x_t, t)\right] \\ &\propto -\mathbb{E}_{t,x_0,x_1}\left[\|v_\theta(x_t, t) - x_1 + x_0\|_2^2\right]. \end{aligned} \quad (3)$$

Note that this objective is identical to classic rectified flow matching. Moreover, note our use of the standard rectified flow velocity vector-field, also derived in Claim 1.

This derivation highlights a key point: because the vector field is parameterized via a Gaussian at each data-time

space location, multi-modality cannot be captured: the Gaussian distribution is uni-modal. Hence, classic rectified flow matching averages the "ground-truth" velocities.

As mentioned before, this can be sub-optimal. To capture multi-modality, we study the use of a mixture model over velocities at each data-time space location. For this, we assume an *unobserved* continuous random variable $z$, drawn from a prior distribution $p(z)$, governs the mean of the *conditional* distribution of the velocity vector-field, i.e.,

$$p(v|x_t, t, z) = \mathcal{N}(v; v_\theta(x_t, t, z), I).$$

Note, this model captures multi-modality as $p(v|x_t, t) = \int p(v|x_t, t, z)p(z)dz$ is a Gaussian mixture.

We now derive the variational flow matching objective. Since the random variable $z$ is not observed, at training time, we introduce a recognition model $q_\phi(z|x_0, x_1, x_t, t)$ a.k.a. an encoder. It is parameterized by $\phi$ and approximates the intractable true posterior.

Using this setup, the marginal likelihood of an individual data point can be lower-bounded by

$$\log p(v|x_t, t) \geq \mathbb{E}_{z \sim q_\phi} \left[ \log p(v|x_t, t, z) \right] \\ - D_{\text{KL}}(q_\phi(\cdot|x_0, x_1, x_t, t)|p(\cdot)). \tag{4}$$

Replacing the log-probability of the Gaussian in the derivation of Equation (3) with the lower bound given in Equation (4) immediately leads to the variational rectified flow matching objective $\mathbb{E}_{t, x_0, x_1} \left[ \log p(x_1 - x_0|x_t, t) \right] \geq$

$$\mathbb{E}_{t, x_0, x_1} [-\mathbb{E}_{z \sim q_\phi} \left[ \|v_\theta(x_t, t, z) - x_1 + x_0\|_2^2 \right] \\ - D_{\text{KL}}(q_\phi(\cdot|x_0, x_1, x_t, t)|p(\cdot))]. \tag{5}$$

We note that this objective could be extended in a number of ways: for instance, the prior $p(z)$ could be a trainable deep net conditioned on $x_0$ and/or $t$. Note however that this leads to a more complex optimization problem with a moving target. We leave a study of extensions to future work.

In Appendix A, we provide a theoretical proof demonstrating that the distribution learned by the variational objective preserves the marginal data distribution, as previously established for classic rectified flow matching (Liu et al., 2023).

In the following we first discuss optimization of this objective before detailing the inference procedure.

### 3.2. Training

To optimize the objective given in Equation (5), we follow the classic VAE setup. Specifically, we let the prior $p(z) = \mathcal{N}(z; 0, I)$ and we let the approximate posterior $q_\phi(z|x_0, x_1, x_t, t) = \mathcal{N}(z; \mu_\phi(x_0, x_1, x_t, t), \sigma_\phi(x_0, x_1, x_t, t))$. This enables analytic computation of the KL-divergence in Equation (5).

---

**Algorithm 1** Variational Rectified Flow Matching Training

**Data:** source distribution $p_0$ and target sample dataset $\mathcal{D}$
**while** stopping conditions not satisfied **do**
    sample $x_0 \sim p_0, x_1 \in \mathcal{D}$ {we use a mini-batch}
    sample $t \sim U(0, 1)$ {different $t$ for each mini-batch sample}
    $x_t = (1 - t)x_0 + tx_1$
    get latent $z = \mu_\phi(x_0, x_1, x_t, t) + \epsilon\sigma_\phi(x_0, x_1, x_t, t)$
    with $\epsilon \sim \mathcal{N}(0, 1)$ {reparameterization trick}
    compute loss following Equation (5)
    perform gradient update on $\theta, \phi$
**end while**

---

**Algorithm 2** Variational Rectified Flow Matching Inference

**Data:** source distribution $p_0$
sample $x_0 \sim p_0$
get latent $z \sim p(z)$
ODE integrate $x_0$ from $t = 0$ to $t = 1$ using velocity vector-field $v_\theta(x_t, t, z)$

---

Note that the mean of the approximate posterior is obtained from the deep net $\mu_\phi(x_0, x_1, x_t, t)$ and the standard deviation is obtained from $\sigma_\phi(x_0, x_1, x_t, t)$. Further, we use the re-parameterization trick to enable optimization of the objective w.r.t. the trainable parameters $\theta$ and $\phi$. Moreover, we use a single-sample estimate for the expectation over the unobserved variable $z$. We summarize the training procedure in Algorithm 1. Note, it's more effective to work with a mini-batch of samples rather than a single data point, which was merely used for readability in Algorithm 1.

Note that variational rectified flow matching training differs from training of classic rectified flow matching in only a single step: computation of a latent sample $z$ in Algorithm 1. From a computational point of view we add a deep net forward pass to obtain the mean $\mu_\phi$ and standard deviation $\sigma_\phi$ of the approximate posterior, and a backward pass to obtain the gradient w.r.t. $\phi$. Also note that the velocity vector-field architecture $v_\theta(x_t, t, z)$ might be more complex as the latent variable $z$ needs to be considered. However, the additional amount of computation is likely not prohibitive.

We provide implementation details for the deep nets $v_\theta(x_t, t, z)$, $\mu_\phi(x_0, x_1, x_t, t)$, and $\sigma_\phi(x_0, x_1, x_t, t)$ in Section 4, as their architecture depends on the data.

### 3.3. Inference

We summarize the inference procedure in Algorithm 2. Note that we sample a latent variable only once prior to classic ODE integration of a random sample $x_0 \sim p_0$ drawn from the source distribution $p_0$. To obtain the latent $z$ we sample from the prior $z \sim p(z) = \mathcal{N}(z; 0, I)$. Subsequently, we ODE integrate the velocity field $v_\theta(x_t, t, z)$ from time $t = 0$ to time $t = 1$ starting from $x_0$.

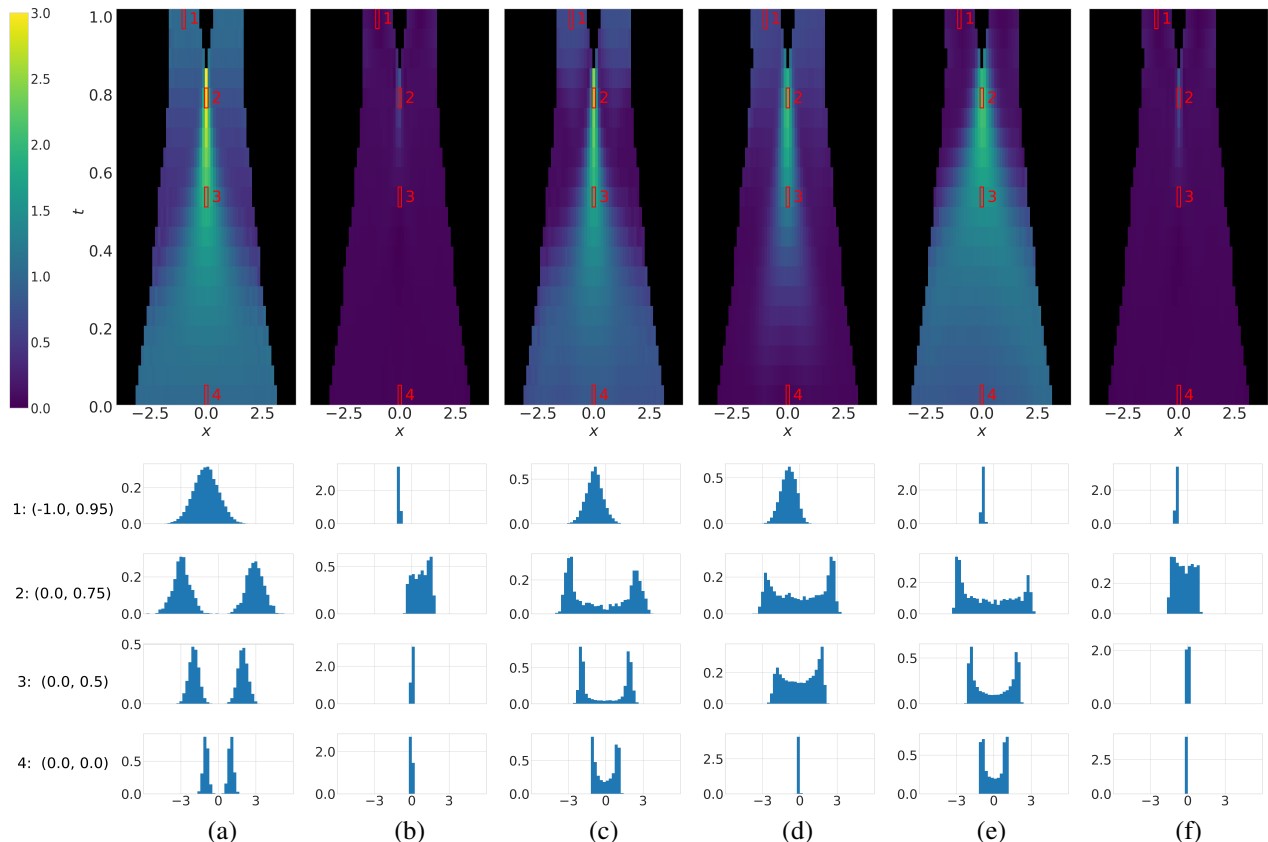

Figure 2: 1D velocity ambiguity analysis with various conditioning options and sampling strategies. (a) Ground Truth (GT), (b) Baseline (Rectified Flow), (c) Ours ($x_0 + x_1 + x_t$), (d) Ours ($x_0$), (e) Ours ($x_1$), (f) Ours ($x_t$). The heatmap illustrates the velocity standard deviation for sampled bins in data-domain-time-domain, along with histograms of the velocity at four sampled locations. Our method effectively models velocity ambiguity, while the baseline produces deterministic outputs.

# 4. Experiments

We evaluate the efficacy of variational rectified flow matching and compare to the classic rectified flow (Lipman et al., 2023; Liu et al., 2023; Albergo & Vanden-Eijnden, 2023) across multiple datasets and model architectures. Our experiments show that variational rectified flow matching is able to capture the multi-modal velocity in the data-time space, leading to compelling evaluation results. Moreover, we demonstrate that explicitly modeling multi-modality through a conditional latent $z$ can enhance the interpretability of flow matching models, leading to controllability. Implementation details for all experiments are provided in Appendix D.

## 4.1. Synthetic 1D Data

For synthetic 1D experiments, the source distribution is a zero-mean, unit-variance Gaussian, while the target distribution is bimodal, with modes centered at $-1.0$ and $1.0$.

For the rectified flow baseline, we use a multi-layer MLP network $v_\theta$ to model the velocity. The network operates on inputs $x_t$ and $t$ and predicts the velocity through a series

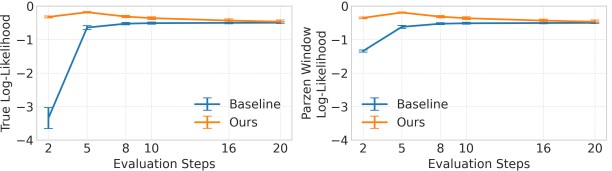

Figure 3: Quantitative evaluation on synthetic 1D data for varying evaluation steps. Metrics are averaged over three runs. For True and Parzen Window Log-Likelihood, higher values are better.

of MLP layers. We follow this structure in our variational rectified flow matching, but add an encoding layer for the latent variable $z$. The posterior model $q_\phi$ follows a similar design as $v_\theta$, outputting $\mu_\phi$ and $\sigma_\phi$. At inference time, $q_\phi$ isn't used. Instead, we sample directly from the prior distribution $p(z) = \mathcal{N}(z; 0, I)$. The KL loss weight is 1.0.

We assess the performance using the Euler ODE solver and vary the evaluation steps. Results are presented in Figure 3. Across both metrics, i.e., True Log-Likelihood and Parzen Window Log-Likelihood, and most evaluation steps, our method outperforms the baseline. Notably, as

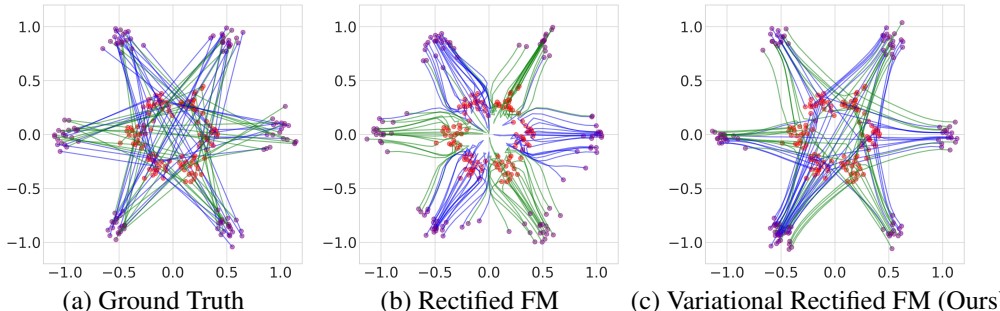

(a) Ground Truth      (b) Rectified FM      (c) Variational Rectified FM (Ours)

Figure 4: Flow visualization for synthetic 2D data using the Euler solver with 20 function evaluations. Sampled points from the source distribution are shown in red, and points from the target distribution in purple. Different from Rectified FM, which predicts flow trajectories with sharp curvature and U-turns to avoid crossings, our model captures velocity ambiguity and predicts flows that intersect.

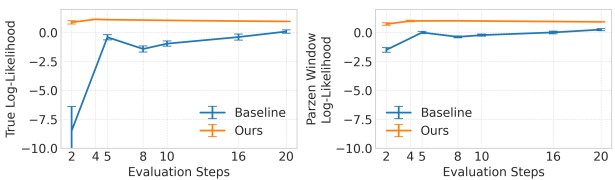

Figure 5: Quantitative evaluation on synthetic 2D data for varying evaluation steps. Metrics are averaged over three runs with different random seeds.

|  | mean curvature | max curvature |
|---|---|---|
| Rectified FM | 21.03 | 171.35 |
| Variational Rectified FM | **0.98** | **4.23** |

Table 1: We calculated the curvature for 2D data results and find significantly lower curvature for our method.

the model handles multi-modality in the data-time space, it produces reasonable results even for 2 or 5 evaluation steps. Qualitative visualizations of flow trajectories are provided in Appendix C.2.

To better understand the multi-modality of the velocity and to assess the efficacy of our model in handling it, we randomly sample different trajectories and plot the velocity range standard deviation across predefined bins in the data-time space, as shown in Figure 2. The ground-truth flow in Figure 2(a) shows that the standard deviation increases with time, peaking at $(x = 0.0, t = 0.75)$. The velocity distribution transitions from a bi-modal distribution at early times $t$ to a uni-modal distribution at later times $t$. Figure 2(b) shows that the rectified flow baseline, which uses an MSE loss, fails to model the velocity distribution faithfully, collapsing to a Dirac-delta distribution as expected. In contrast, Figure 2(c) demonstrates that our model captures the distribution with higher velocity standard deviation range, matching the ground-truth reasonably, albeit not perfectly.

As discussed in Section 3.2, the posterior $q_\phi$ can be conditioned in different ways. To understand the implications, we performed ablation studies and visualized the velocity distribution maps in Figure 2 (c)-(f). For $x_0$ conditioning (d), the model struggles to predict the bi-modal distribution at early timesteps $(x_t = 0.0, t = 0.0)$ due to the absence of $x_1$ information. However, when $t$ is sufficiently large, the model can infer $x_1$ from $x_t$, enabling it to predict a bimodal distribution again at $(x = 0.0, t = 0.5)$. Conversely, with

$x_1$ conditioning (e), the model fails to capture the ground-truth distribution at later timesteps $(x = -1.0, t = 0.95)$ as the influence of $x_1$ diminishes. With $x_t$ conditioning (f), the ambiguity plot follows the baseline as no extra data is provided to the posterior.

### 4.2. Synthetic 2D Data

We further test efficacy using synthetic 2D data. Following Liu et al. (2023), we model the source distribution as a mixture of Gaussian components positioned at six equidistant points along a circle with a radius of 1/3, shown in red in Figure 4(a). The target distribution follows a similar structure, but with components arranged along a larger circle with a radius of 1, shown in purple.

For the architecture we follow Section 4.1 and condition the posterior on $[x_0, x_1, x_t]$. We report the True Log-Likelihood and the Parzen Window Log-Likelihood for various evaluation steps of the Euler ODE solver, as shown in Figure 5. Compared to the 1D data, our model shows a more significant performance boost here. We hypothesize that this is due to the increased complexity of the task: explicitly modeling multi-modality avoids the need for curved trajectories, making it easier to fit the target distribution. The qualitative flow visualization in Figure 4 supports this hypothesis: the rectified flow requires a U-turn to avoid collisions, while our model, aided by the variational training objective, moves in trajectories that intersect and aren't as curved. To verify, we calculated the curvature of flow trajectories for both the baseline and our variational rectified flow matching method, and report the results in Table 1. Our method yields significantly lower curvature, indicating smoother flow trajectories.

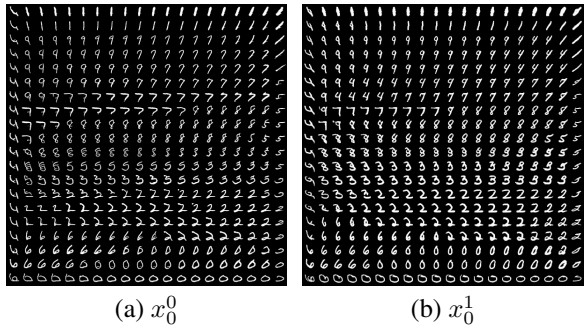

(a) $x_0^0$        (b) $x_0^1$

Figure 6: Visualization of learned MNIST manifold with different random noise $x_0$.

| NFE | 2 | 5 | 10 | 20 | 50 | 100 | 200 |
|---|---|---|---|---|---|---|---|
| Baseline | 98.8 | 26.9 | 13.0 | 8.8 | 7.0 | 6.4 | 6.0 |
| Ours | **61.0** | **13.4** | **6.8** | **5.1** | **4.7** | **4.7** | **4.6** |

Table 2: FID score evaluation for the MNIST experiment conducted across three trials with different random seeds. Our model with a latent dimension of 2 outperforms the baseline. Note, the latent dimension of 2 is chosen for a controllability analysis rather than being optimized for FID score improvement.

### 4.3. MNIST

Modeling multi-modality also enables more explicit control without additional conditioning signals. To show this we use variational rectified flow matching to train a vanilla convolutional net with residual blocks (He et al., 2015) on MNIST data (LeCun et al., 1998). We use $(x_0, x_1, x_t)$ as input to $q_\phi$ and set the KL loss weight to $1e^{-3}$.

Following Kingma & Welling (2014), we set the latent variable $z$ to be 2-dimensional. During inference, we sample linearly spaced coordinates on the unit square, transforming them through the inverse CDF of the Gaussian to generate latents $z$. Using these latents, we integrate the samples with an ODE solver and plot the generated samples in Figure 6. To show the effects of the source distribution sample $x_0$ and the latent $z$, we visualize the learned MNIST manifold for two randomly sampled $x_0$ values in Figure 6(a,b). The results demonstrate that the latent space $z$ enables smooth interpolation between different digits within the 2D manifold, providing control over the generated images. By adjusting $z$, we can transition between various shapes and styles. The initial noise $x_0$ enhances the generation process by introducing additional variations in character styles, allowing the model to better capture the target data distribution. We also evaluate the FID scores of our method using this 2-dimensional conditional latent space and report the results in Table 2. Despite the small latent dimension, it still enables the velocity model $v_\theta$ to achieve better FID scores than the rectified flow matching baseline.

Our method achieves better FID scores than the baseline, even with a 2-dimensional conditional latent space. Further details and results can be found in Appendix C.3.

### 4.4. CIFAR-10

Next, we evaluate on CIFAR-10, a widely used benchmark in prior work (Lipman et al., 2023; Tong et al., 2024). For a fair comparison, we use the architecture and training paradigm of Tong et al. (2024), but train the UNet model with the variational rectified flow loss detailed in Equation (5). The UNet consists of downsampling and upsampling residual blocks with skip connections, and a self-attention block added after the residual block at $16 \times 16$ resolution and in the middle bottleneck layer. The model takes both $x_t$ and $t$ as input, with the time embedding $t$ used to regress learnable scale and shift parameters $\gamma$ and $\beta$ for adaptive group norm layers.

The posterior model $q_\phi$ shares a similar encoder structure as $v_\theta$: image space inputs are chosen from $[x_0, x_1, x_t]$ and concatenated along the channel dimension, while time $t$ is conditioned using adaptive group normalization. The network predicts $\mu_\phi$ and $\sigma_\phi$ with dimensions $1 \times 1 \times 768$. During training, the conditional latent $z$ is sampled from the predicted posterior, and at test time, from a standard Gaussian prior. The latent is processed through two MLP layers and serves as a conditional signal for the velocity network $v_\theta$. We identify two effective approaches as conditioning mechanisms: adaptive normalization, where $z$ is added to the time embedding before computing shift and offset parameters, and bottleneck sum, which fuses the latent with intermediate activations at the lowest resolution using a weighted sum before upsampling.

We evaluate results using FID scores computed for varying numbers of function evaluations, as shown in Table 3. Four model variants were tested, differing in fusion mechanisms, posterior model $q_\phi$ inputs, and KL loss weighting. Compared to prior work (Lipman et al., 2023; Liu et al., 2023; Tong et al., 2024), model 1 achieves superior FID scores with fewer function evaluations and performs comparably at higher evaluations. Using the adaptive Dopri5 solver further improves scores, highlighting the importance of capturing flow ambiguity. Model 2 increases the KL loss weight, improving performance at higher function evaluations but reducing effectiveness at lower evaluations, likely due to reduced information from latent $z$. Model 3, with additional time conditioning, significantly improves FID at low evaluations and performs best with the adaptive solver. Model 4, incorporating bottleneck sum fusion, delivers robust FID scores across evaluation settings, demonstrating the flexibility of the variational rectified flow objective with different fusion strategies.

Similar to the MNIST results in Section 4.3, we observe

| | NFE / sample | # Params. | 2 | 5 | 10 | 50 | 100 | 1000 | Adaptive |
|---|---|---|---|---|---|---|---|---|---|
| | I-CFM (Tong et al., 2024) | 36.5M | 168.654 | 35.489 | 13.788 | 5.288 | 4.461 | 3.643 | 3.659 |
| 1 | V-RFM (adaptive norm, $x_1$, 2e-3) | 37.2M | 135.275 | 28.912 | **13.226** | 5.382 | 4.430 | 3.642 | 3.545 |
| 2 | V-RFM (adaptive norm, $x_1$, 5e-3) | 37.2M | 159.940 | 35.293 | 14.061 | **5.265** | **4.349** | **3.582** | 3.561 |
| 3 | V-RFM (adaptive norm, $x_1 + t$, 5e-3) | 37.2M | 117.666 | 27.464 | 13.632 | 5.512 | 4.484 | 3.614 | **3.478** |
| 4 | V-RFM (bottleneck sum, $x_1 + t$, 2e-3) | 37.0M | **104.634** | **25.841** | 13.508 | 5.618 | 4.540 | 3.596 | 3.520 |

Table 3: Following Tong et al. (2024), we train the same UNet model and reported the FID scores for our method and the baselines using both fixed-step Euler and adaptive-step Dopri5 ODE solvers. The baselines checkpoint was directly taken from Tong et al. (2024). We present four model variants of our V-RFM, which differ in fusion mechanism, posterior model input, and KL loss weight.

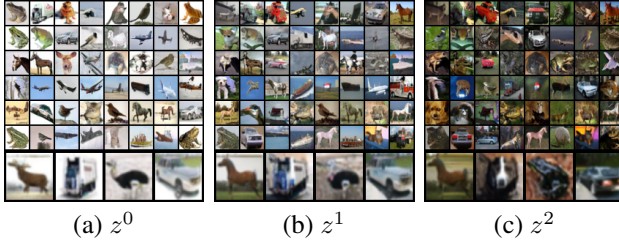

| (a) $z^0$ | (b) $z^1$ | (c) $z^2$ |

Figure 7: Varying $x_0$ while keeping the latent $z$ fixed. Images at the same position across panels share the same $x_0$, while images within a panel share the same latent sampled from the prior distribution.

| Model | Params (M) | Training Steps | FID ↓ |
|---|---|---|---|
| DiT-XL | 675 | 400K | 19.5 |
| SiT-XL | 675 | 400K | 17.2 |
| **V-SiT-XL** | 677 | 400K | **14.6** |
| SiT-XL | 675 | 800K | 13.1 |
| **V-SiT-XL** | 677 | 800K | **10.6** |
| SiT-XL_{cfg=1.5} | 675 | 400K | 5.40 |
| **V-SiT-XL**_{cfg=1.5} | 677 | 400K | **4.91** |
| SiT-XL_{cfg=1.5} | 675 | 800K | 3.43 |
| **V-SiT-XL**_{cfg=1.5} | 677 | 800K | **3.22** |

Table 4: FID-50K score evaluation of class-conditional generation on ImageNet $256 \times 256$, comparing the baselines (DiT-XL, SiT-XL) with our proposed model **V-SiT-XL**.

clear patterns in color and content for the generated samples $x_1$, demonstrating a degree of controllability. Figure 7 visualizes three sets of images (a)–(c). Each set is conditioned on a different latent $z$, while the starting noise $x_0$ varies across individual images within each set. The same noise $x_0$ is applied to images at the same grid location across all subplots. Images conditioned on the same latent exhibit consistent color patterns, while images at the same grid location display similar content, as highlighted in the last row.

### 4.5. ImageNet

To assess efficacy on large-scale data, we use ImageNet $256 \times 256$ data and SiT-XL (Ma et al., 2024), a recent transformer-based model that has shown strong results in image generation. For a fair comparison, we strictly follow the original training recipe in the open-source SiT repository and replicate the training process from the SiT paper, while introducing our model, V-SiT-XL, by substituting the classic rectified flow loss with the variational rectified flow loss in Equation (5). The posterior model $q_\phi$ also utilizes an SiT transformer architecture but with half the number of blocks. In the final layer, the features are average pooled and passed through an MLP layer to predict $\mu_\phi$ and $\sigma_\phi$. We sample the latent variable $z$ from the posterior during training and from the prior distribution during inference. This latent variable $z$ is then processed by two MLP layers and fused with the velocity network $v_\theta$ via adaptive normalization. By default,

we use the Euler-Maruyama sampler with the SDE solver and 250 integration steps, as described by Ma et al. (2024).

Following the evaluation protocol of Ma et al. (2024), we randomly generate 50K images from the models and report the FID scores in Table 4. V-SiT-XL consistently outperforms both DiT-XL and SiT-XL, achieving gains under the same training conditions, with and without classifier-free guidance. These results underscore the importance of modeling multi-modality in the velocity vector field, which contributes to a substantial improvement in generation quality, particularly in the large-scale high-resolution data domain. Additionally, we analyze the model's performance across different training iterations and varying numbers of function evaluations, presenting the findings in Figure 8. The results reveal a consistent performance boost, further highlighting the effectiveness of our approach.

## 5. Related Work

Generative modeling has advanced significantly in the last decade, thanks in part due to seminal works like generative adversarial nets (Goodfellow et al., 2014), variational auto-encoders (Kingma & Welling, 2014), and normalizing flows (Rezende & Mohamed, 2015).

More recently, score matching (Song & Ermon, 2019) and

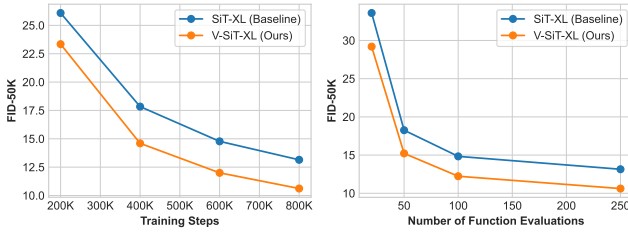

Figure 8: FID-50K score over training iterations and number of function evaluations. Our model, V-SiT-XL, consistently achieves a better FID score compared to SiT-XL trained with classic rectified flow matching.

diffusion models (Ho et al., 2020) were introduced. They can be viewed as augmenting variational auto-encoders hierarchically (Luo, 2022) while restricting involved distributions to be Gaussian. Notably, and analogously to classic discrete normalizing flows, the number of hierarchy levels, i.e., the number of time steps, remained discrete, which introduced complications.

Flow matching (Lipman et al., 2023) was introduced recently as a compelling alternative to avoid some of these complications. It formulates an ordinary differential equation (ODE) in continuous time. This ODE connects a source distribution to a target distribution. Solving the ODE via forward integration through time permits to obtain samples from the target distribution, essentially by 'moving' samples from the known source distribution to the target time along a learned velocity field.

To learn the velocity field, various mechanisms to interpolate between the source distribution and the target distribution have been considered (Lipman et al., 2023; Liu et al., 2023; Tong et al., 2024). Rectified flow matching emerged as a compelling variant, which linearly interpolates between samples from the two distributions. For instance, it was used to attain impressive results on large scale data (Ma et al., 2024; Esser et al., 2024). Different from other techniques, linear interpolation encourages somewhat straight flows, which simplifies numerical solving of the ODE.

The importance of straight flows was further studied in ReFlow (Liu et al., 2023), which sequentially formulates multiple ODEs and learns velocity fields by adjusting the interpolations and 're-training.' Consistency models (Song et al., 2023; Kim et al., 2023; Yang et al., 2024) strive for straight flows by modifying the loss to encourage self-consistency across timesteps. More details and comparisons are provided in Appendix B and C.1.

While the aforementioned works aim to establish straight flows either via 're-training' or 're-parameterizing' of an already existing flow, differently, in this work we study the results of enabling a rectified flow to capture the ambiguity inherent in the usually employed ground-truth flow fields.

Structurally similar to this idea is work by Preechakul et al. (2022). In a first stage, an autoencoder is trained to compress images into a latent space. The resulting latents then serve as a conditioning signal for diffusion model training in a second stage. Note, this two-stage approach doesn't directly model ambiguity in the data-time space. In spirit similar is work by Pandey et al. (2022). A VAE and a diffusion model are trained in two separate stages, with the goal to enable controllability of diffusion models. Related is also work by Eijkelboom et al. (2024) which focuses on flow matching only for categorical data, achieving compelling results on graph generation tasks. Another related work by Vahdat et al. (2021) uses a VAE to map raw data $x_0$ into latent space $z_0$, with the VAE jointly trained with score-based generative modeling (SGM). Unlike our approach, SGM still faces ambiguity issues due to its use of a uni-modal Gaussian distribution.

Related work by Xiao et al. (2022) replaces the Gaussian model in the denoising step with a multimodal distribution. Unlike our method, a conditional GAN with separate discriminator models the distribution. But GANs face mode collapse and stability issues. In contrast, our method uses rectified flow matching, preserving the maximum likelihood benefits. Xu et al. (2023) notes that the posterior distribution is multi-modal. To model this distribution, the paper reduces training target variance using a reference batch. In contrast, our method directly models this multi-modal posterior via a recognition model.

Concurrently, Zhang et al. (2025) also study a method to model multi-modal velocity vector fields. In this paper, we discuss how to use a lower-dimensional latent space to enable modeling of the velocity distribution via a variational approach. Differently, Zhang et al. (2025) study use of a hierarchy of ordinary differential equations. The variational approach enables to capture semantics while use of a hierarchy of ordinary differential equations permits to more accurately model the velocity distribution.

## 6. Conclusion

We study Variational Rectified Flow Matching, a framework which enables to model the multi-modal velocity vector fields induced by the ground-truth linear interpolation between source and target distribution samples. Encouraging results can be obtained on low-dimensional synthetic and high-dimensional image data.

**Acknowledgements:** We thank Zhizhen Zhao for insightful feedback.

## Impact Statement

This paper presents work whose goal is to advance the field of machine learning. There are many potential societal consequences of our work, none which we feel must be specifically highlighted here.

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

# Appendix: Variational Rectified Flow Matching

This appendix is structured as follows: in Appendix A we show that our approach maintains the marginal distribution; in Appendix B we discuss additional related work; in Appendix C we provide additional experimental analysis; in Appendix D we provide more implementation details; in Appendix E we list additional qualitative results.

## A. On Preserving the Marginal Data Distribution

We obtain samples by numerically solving the ordinary differential equation

$$du_t = v_\theta(x_t, t, z)dt \quad \text{with} \quad z \sim p(z) = \mathcal{N}(z; 0, I).$$

This differs slightly from Theorem 3.3 of Liu et al. (2023) because the velocity $v_\theta$ depends on a latent variable $z$ drawn from a standard Gaussian. However, Theorem 3.3 of Liu et al. (2023) can be extended to fit this setting as follows.

First, note that we have $v^*(x_t, t, z) = \mathbb{E}[\dot{X}_t | X_t, Z]$ where $X_t$ and $Z$ are random variables corresponding to instances $x_t$ and $z$.

Incorporating the velocity field depending on the latent variable $z$ into the transport problem defined in Equation (2) and taking an expectation over the latent variable, we obtain the continuity equation

$$\dot{p}_t + \text{div}(\mathbb{E}_Z[v_\theta(x_t, t, z)]p_t) = 0. \tag{6}$$

Following Liu et al. (2023), one can show equivalence to the following equality, which uses any compactly supported continuously differentiable test function $h$:

$$\frac{d}{dt}\mathbb{E}[h(X_t)] = \mathbb{E}[\nabla h(X_t)^T \dot{X}_t] = \mathbb{E}[\nabla h(X_t)^T v^*(X_t, t)] = \mathbb{E}_X[\nabla h(X_t)^T \mathbb{E}_Z[v^*(X_t, t, Z)]].$$

Concretely, equivalence can be shown via

$$0 = \mathbb{E}_Z\left(\int_{x_t} h(\dot{p}_t + \text{div}(v^*(X_t, t, Z)p_t)\right) = \frac{d}{dt}\mathbb{E}[h(X_t)] - \mathbb{E}_X[\nabla h(X_t)^T \mathbb{E}_Z[v^*(X_t, t, Z)]].$$

Note, different from Liu et al. (2023), in our case $U_t$ is driven by a velocity field $v(x_t, t, z)$ that depends on a latent variable. Averaging over instantiations of the random latent variable $Z$ leads to the same marginal velocity that appears in the continuity equation (Equation (6)). Therefore, we solve the same equation with the same initial condition ($X_0 = U_0$). Equivalence follows if the solution to Equation (6) is unique.

## B. Additional Related Work Discussion

Here, we discuss related work aimed at improving the sample efficiency of diffusion and flow matching models, either via consistency modeling or via distillation.

**Consistency models.** Consistency models, such as those by Song et al. (2023) and Yang et al. (2024), enforce self-consistency across timesteps, ensuring trajectories map back to the same initial point. Moreover, Kim et al. (2023) ensure consistent trajectories for probability flow ODEs. While consistency models focus on improving results via trajectory alignment if few function evaluations are used, they don't model the multi-modal ground-truth velocity distribution, which is our goal.

To illustrate this, we train the recently developed consistency flow matching model proposed by Yang et al. (2024) (which improves upon work by Song et al. (2023) and Kim et al. (2023); both are not flow matching based; it also improves upon distillation work by Nguyen et al. (2024)) on the data for which V-RFM results are presented in Figures 2 and 13. Specifically, we used the publicly available baseline.[1] We obtain the results illustrated in Figure 9. As expected, we observe

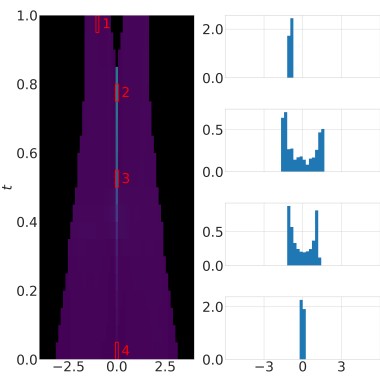

Figure 9: Velocity distribution of consistency flow matching (Yang et al., 2024).

that classic consistency modeling does not capture the multi-modal velocity distribution, unlike the proposed V-RFM.

---

[1]https://github.com/YangLing0818/consistency_flow_matching

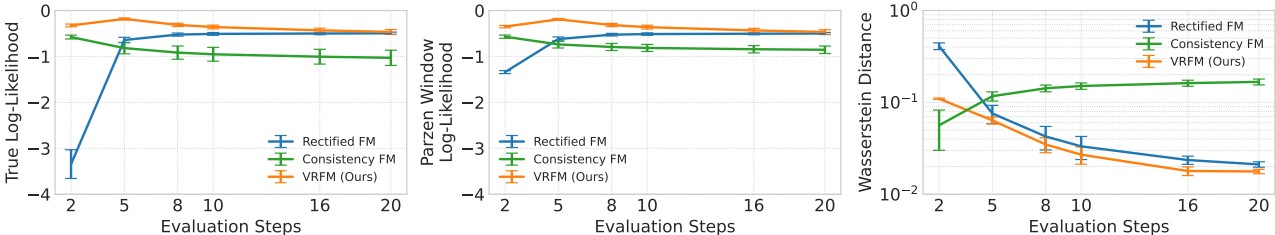

Figure 10: Additional quantitative evaluation with the consistency flow matching baseline on synthetic 1D data. Higher values are better for True and Parzen Window Log-Likelihood, while lower values are preferred for Wasserstein Distance.

Figure 11: Additional quantitative evaluation with the consistency flow matching baseline on synthetic 2D data. Metrics are averaged over three runs with different random seeds.

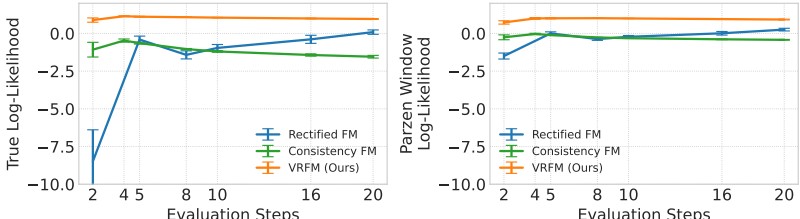

Furthermore, we conduct additional experiments with consistency flow matching across multiple datasets, summarizing the results in Appendix C.1. We observe that the consistency flow matching method performs well in the low function evaluation regime (i.e., NFE = 2 or 5), but its performance degrades as the NFEs increase. Most notably, its best performance across all NFEs does not surpass that of classic rectified flow matching or our proposed variational rectified flow matching. Based on the empirical evidence and the key differences in capturing multi-modal velocity distributions, we believe consistency models are orthogonal to our proposed variational formulation. Therefore, we find it exciting to explore future research on combining variational flow matching with consistency models, which is beyond the scope of this paper.

**Distillation.** Nguyen et al. (2024) perform distillation by optimizing step sizes in pretrained flow-matching models to refine trajectories and improve training dynamics. Moreover, Yan et al. (2024) perform distillation by introduceing a piecewise rectified flow mechanism to accelerate flow-based generative models. Note, both methods distill useful information from a pretrained model, either by using dynamic programming to optimize the step size or by applying reflow to straighten trajectories, i.e., they focus on distilling already learned models. In contrast, our V-RFM focuses on learning via single-stage training, directly from ground-truth data, and without use of a pre-trained deep net, a flow-matching model, which captures a multi-modal velocity distribution. More research on the distillation of a V-RFM model is required to assess how multi-modality can be maintained in the second distillation step. We think this is exciting future research, which is beyond the scope of this paper.

**Reflow.** We have added a comparison with 2/3-Rectified Flow via Reflow (Liu et al., 2023) in Table 5. We find that while strong FID scores in the low-NFE regime are achieved, it does so at the cost of limiting peak performance at high NFE. We emphasize that Reflow is a supplementary technique applied on top of Rectified Flow Matching (RFM)—primarily aimed at fast sampling rather than improved sample quality. It also requires $N$ times longer training and a significantly larger fine-tuning dataset, where $N$ denotes the number of Reflow rounds. These differences make a direct comparison with our V-RFM less fair. Hence, RFM without Reflow is a more appropriate baseline. Additionally, Reflow can be applied to our method as well, potentially improving results at the cost of increased training overhead.

## C. Additional Experimental Results and Analysis

### C.1. Comparison to Consistency Flow Matching

We conduct additional experiments to compare our approach with consistency models across multiple datasets. For this, we use the recently developed consistency flow matching model from Yang et al. (2024) as a representative baseline, as it advances earlier consistency modeling efforts by Song et al. (2023); Kim et al. (2023) and distillation work by Nguyen et al. (2024). Specifically, we used the publicly available implementation.[2]

The results are summarized as follows: Synthetic 1D data in Figure 10, Synthetic 2D data in Figure 11, MNIST data in

---

[2]https://github.com/YangLing0818/consistency_flow_matching

Figure 12: FID score evaluation for the MNIST experiment, including the additional consistency flow matching baseline. Our model with a latent dimension of 2 outperforms the baselines, except at 2 evaluation steps where Consistency FM performs best. Note, the latent dimension of 2 is chosen for a controllability analysis rather than being optimized for FID score improvement.

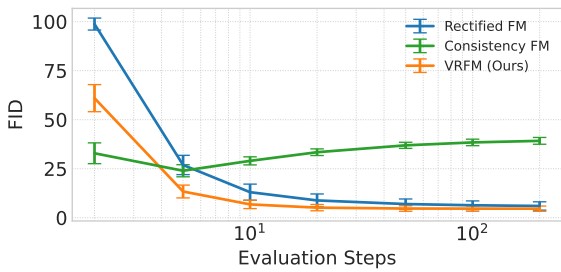

| | NFE / sample | # Params. | 2 | 5 | 10 | 50 | 100 | 1000 | Adaptive |
|---|---|---|---|---|---|---|---|---|---|
| | OT-FM (Lipman et al., 2023; Tong et al., 2024) | 36.5M | 166.655 | 36.188 | 14.396 | 5.557 | 4.640 | 3.822 | 3.655 |
| | I-CFM (Liu et al., 2023; Tong et al., 2024) | 36.5M | 168.654 | 35.489 | 13.788 | _5.288_ | 4.461 | 3.643 | 3.659 |
| | I-CFM w. 1 Reflow (Liu et al., 2023; Tong et al., 2024) | 36.5M | 7.512 | 5.906 | 5.513 | _5.283_ | 5.276 | 5.276 | 5.275 |
| | I-CFM w. 2 Reflow (Liu et al., 2023; Tong et al., 2024) | 36.5M | 7.559 | 6.925 | 6.776 | 6.729 | 6.733 | 6.752 | 6.752 |
| | Consistency-FM (Yang et al., 2024) | 36.5M | _15.758_ | _14.588_ | 24.107 | 36.800 | 38.675 | 40.486 | 40.711 |
| | Consistency-FM-XL (Yang et al., 2024) | 61.8M | **5.323** | **11.412** | 23.948 | 36.652 | 38.680 | 40.402 | 40.677 |
| 1 | V-RFM (adaptive norm, $x_1$, 2e-3) | 37.2M | 135.275 | 28.912 | **13.226** | 5.382 | _4.430_ | 3.642 | 3.545 |
| 2 | V-RFM (adaptive norm, $x_1$, 5e-3) | 37.2M | 159.940 | 35.293 | 14.061 | **5.265** | **4.349** | **3.582** | 3.561 |
| 3 | V-RFM (adaptive norm, $x_1 + t$, 5e-3) | 37.2M | 117.666 | 27.464 | 13.632 | 5.512 | 4.484 | 3.614 | **3.478** |
| 4 | V-RFM (bottleneck sum, $x_1 + t$, 2e-3) | 37.0M | 104.634 | 25.841 | _13.508_ | 5.618 | 4.540 | _3.596_ | _3.520_ |

Table 5: Additional quantitative evaluation of the consistency flow matching baseline on CIFAR-10. The consistency flow matching method performs well in the low function evaluation regime (NFE = 2 or 5), but its performance degrades as NFEs increase. Notably, its best performance across all NFEs does not surpass that of classic rectified flow matching (OT-FM, I-CFM) or our proposed variational rectified flow matching (V-RFM).

Figure 12, and CIFAR-10 data in Table 5. These results demonstrate that V-RFM outperforms the consistency flow matching baseline across various evaluation steps for synthetic data, with V-RFM showing superior performance when the number of evaluation steps exceeds 2 for MNIST and 5 for CIFAR-10. Importantly, while consistency flow matching achieves strong performance for a low number of evaluation steps, its best performance still does not surpass that of classic rectified flow matching or our proposed variational rectified flow matching with a high number of evaluation steps. This highlights its distinct nature as an orthogonal research direction to our method. As discussed in Appendix B, we believe that combining variational formulations with consistency models presents an exciting avenue for future research, though it is beyond the scope of this paper.

### C.2. Qualitative Results of Synthetic 1D Experiment

We provide qualitative flow visualizations from the synthetic 1D experiment in Figure 13. Our method effectively captures velocity ambiguity and predicts crossing flows, whereas the baselines produce deterministic outputs.

### C.3. Quantitative Results of MNIST Experiment

We evaluate the FID scores of our method using this 2-dimensional conditional latent space and report the results in Figure 12. Despite the small latent dimension, it still enables the velocity model $v_\theta$ to achieve better FID scores than the baselines, except at 2 evaluation steps where consistency flow matching (Yang et al., 2024) performs best.

### C.4. Inception Score Evaluation of CIFAR-10 Experiment

We evaluate the Inception Score of our model trained on CIFAR-10 data and present results in Table 6. This score quantifies the distribution of predicted labels for the generated samples. Compared to the vanilla rectified flow baseline, our method

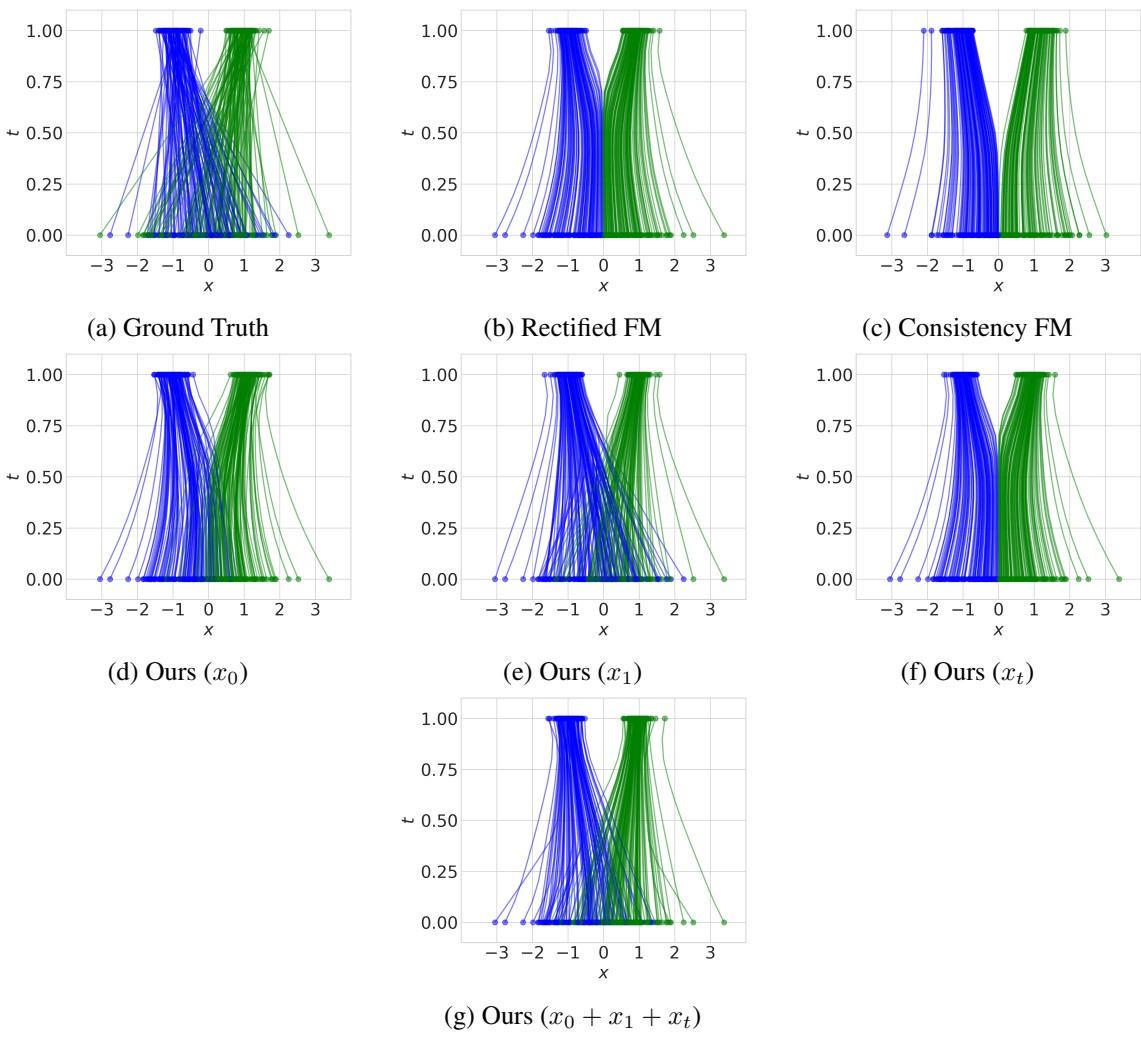

(a) Ground Truth     (b) Rectified FM     (c) Consistency FM

(d) Ours ($x_0$)     (e) Ours ($x_1$)     (f) Ours ($x_t$)

(g) Ours ($x_0 + x_1 + x_t$)

Figure 13: 1D flow visualization for uni-modal Gaussian to bi-modal Gaussian.

consistently achieves higher Inception Scores, reflecting improved diversity in the generated samples.

### C.5. Ablation on Posterior Model Size

We conducted ablations to study the impact of varying the size of the encoder $q_\phi$, reducing it to 6.7% and 17.5% of its original size. The results reported in Table 7 demonstrate that our model maintains comparable performance across these variations, highlighting the flexibility and robustness of our approach.

### C.6. Reconstruction Loss Visualizations

We present the reconstruction loss curves for our model and the baseline trained on MNIST and CIFAR-10 data in Figure 14. We observe better reconstruction losses of our model compared to vanilla rectified flow, indicating that the predicted velocities more accurately approximate the ground-truth velocities.

## D. Implementation Details

### D.1. Synthetic data

In the rectified flow baseline, the velocity network $v_\theta$ features separate encoders for time $t$ and data $x$. Each encoder consists of a sinusoidal positional encoding layer followed by two MLP layers with GeLU activation. The resulting time and data embeddings are concatenated and passed into a four-layer MLP, also utilizing GeLU activations. Both the positional

| NFE / sample | | 2 | 5 | 10 | 50 | 100 | 1000 | Adaptive |
|---|---|---|---|---|---|---|---|---|
| | I-CFM (Liu et al., 2023; Tong et al., 2024) | 2.786 | 7.143 | 8.326 | 8.770 | 8.872 | 9.022 | 9.041 |
| 1 | V-RFM (adaptive norm, $x_1$, 2e-3) | 3.943 | 7.728 | 8.499 | 8.973 | 9.050 | 9.168 | 9.171 |
| 2 | V-RFM (adaptive norm, $x_1$, 5e-3) | 3.083 | 7.202 | 8.342 | 8.868 | 8.997 | 9.166 | 9.183 |
| 3 | V-RFM (adaptive norm, $x_1 + t$, 5e-3) | 4.460 | 7.930 | **8.583** | 9.007 | 9.104 | 9.220 | 9.238 |
| 3 | V-RFM (bottleneck sum, $x_1 + t$, 2e-3) | **4.831** | **7.996** | 8.529 | **9.062** | **9.150** | **9.293** | **9.308** |

Table 6: Inception Score evaluation of our method compared to the baseline on CIFAR-10, using fixed-step Euler and adaptive-step Dopri5 ODE solvers. Higher scores indicate better performance.

| NFE / sample | | 2 | 5 | 10 | 50 | 100 | 1000 | Adaptive |
|---|---|---|---|---|---|---|---|---|
| | OT-FM (Lipman et al., 2023; Tong et al., 2024) | 166.655 | 36.188 | 14.396 | 5.557 | 4.640 | 3.822 | 3.655 |
| | I-CFM (Liu et al., 2023; Tong et al., 2024) | 168.654 | 35.489 | 13.788 | **5.288** | 4.461 | 3.643 | 3.659 |
| 1 | V-RFM-L (100% Posterior Model) | **135.275** | **28.912** | **13.226** | 5.382 | 4.430 | **3.642** | **3.545** |
| 2 | V-RFM-M (17.5% Posterior Model) | 135.983 | 30.106 | 13.783 | 5.486 | 4.500 | 3.697 | 3.607 |
| 3 | V-RFM-S (6.7% Posterior Model) | 144.676 | 31.224 | 13.406 | 5.289 | **4.398** | 3.699 | 3.639 |

Table 7: We use the same flow matching model $v_\theta$ and pair it with different sizes of encoders $q_\phi$ during training while maintaining the exact same hyper-parameters. We report the FID scores for our method and the baseline using both fixed-step Euler and adaptive-step Dopri5 ODE solvers.

embedding and hidden dimensions of the encoder and decoder are set to 64. The training batch size is 1000, and we employ the standard rectified flow objective, i.e., we compute the current data via $x_t = (1 - t)x_0 + tx_1$, the ground truth velocity via $v(x_0, x_1, t) = x_1 - x_0$, and we use the L2 loss for supervision.

For consistency flow matching, we adopt the same velocity network $v_\theta$ and modify the loss function to incorporate the velocity consistency loss proposed by Yang et al. (2024). We find the hyperparameter settings suggested by the publicly available codebase to work best. Specifically, we use $\Delta t = 1 \times 10^{-3}$, $N_{segments} = 2$, and *boundary* = 0.0 for the first training stage, transitioning to *boundary* = 0.9 in the second stage. Additionally, the loss weighting factor $\alpha$ is set to $1 \times 10^{-5}$. For complete implementation details, we kindly direct readers to the open-source repository which we used to obtain the reported results.[3]

In both cases, the AdamW optimizer is used with the default weight decay and a learning rate of $1 \times 10^{-3}$, over a total of 20,000 training iterations.

In our variational flow matching approach, the velocity network $v_\theta$ incorporates an additional latent encoding module comprising three MLP layers with a hidden dimension of 128. The conditional latent embedding $z$ is concatenated with the embeddings for time $t$ and data $x$. The decoder maintains the same structure as the baseline, with the first MLP layer adjusted to accommodate the increased channel input. For the posterior model $q_\phi$, we employ a similar architecture, designing a separate encoder for each possible input selected from $[x_0, x_1, x_t, t]$. Each encoder consists of a sinusoidal positional encoder layer followed by two MLP layers with GeLU activation. The output embeddings are concatenated along the channel dimension and processed through three MLP layers to produce the predicted $\mu_\phi$ and $\sigma_\phi$. The latent dimension of $z$ is set to 4 for 1D experiments and 8 for 2D experiments. During training, we utilize the reparameterization trick to sample $z$ from the predicted posterior distribution; during inference, the posterior model $q_\phi$ is omitted, and sampling is performed from a unit variance Gaussian prior distribution. The loss is defined as the sum of the rectified flow reconstruction loss and the KL divergence loss, with the KL loss weighted at 1.0 for the 1D experiments and 0.1 for the 2D experiments. We employ AdamW as the optimizer with a learning rate of $1 \times 10^{-3}$ and train the two networks $q_\phi$ and $v_\theta$ jointly for 20,000 iterations.

### D.2. MNIST

In the rectified flow baseline, the velocity network $v_\theta$ uses separate encoders for time $t$ and data $x$. The time $t$ encoder consists of a sinusoidal positional encoding layer followed by two MLP layers with SiLU activation. The data $x$ encoder includes a convolutional in-projection layer, five consecutive ResNet (He et al., 2015) blocks (each consisting of two

---

[3]https://github.com/YangLing0818/consistency_flow_matching

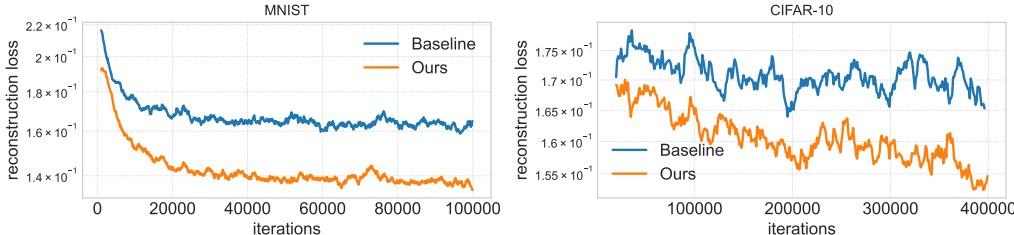

Figure 14: Reconstruction loss for MNIST (left) and CIFAR-10 (right). We observe lower reconstruction losses for the variational formulation, indicating a better fit.

convolutional layers with a kernel size of 3, group normalization, and SiLU activation), followed by a convolutional out-projection layer. The time and data embeddings are concatenated and passed to a decoder composed of a convolutional in-projection layer, five consecutive ResNet blocks, and a convolutional out-projection layer with a kernel size of 1 and an output channel of 1. The hidden dimension is set to 64. MNIST data is normalized to the $[-1, 1]$ range. We adopted the consistency velocity loss from the consistency flow matching baseline used for synthetic data experiments. We train the network for 100,000 iterations using the AdamW optimizer with a learning rate of $1 \times 10^{-3}$ and batch size of 256.

In our variational flow matching approach, the velocity network $v_\theta$ includes an additional latent encoding module consisting of a sinusoidal positional encoding layer followed by two MLP layers with SiLU activation. The conditional latent embedding $z$ is concatenated with the embeddings for time $t$ and data $x$. The decoder structure mirrors the baseline, with the first in-projection layer adjusted to handle the increased channel input. The posterior model $q_\phi$ follows a similar architecture, with separate encoders for each input $[x_0, x_1, x_t]$. The resulting embeddings are concatenated and passed through a decoder consisting of a convolutional in-projection layer, followed by three consecutive interleaving ResNet blocks and average pooling layers. The final hidden activation is flattened and processed by two linear MLP layers to predict the 1D latent $z$ with a dimension of 2. The two networks are trained jointly for 100,000 iterations using the AdamW optimizer with a learning rate of $1 \times 10^{-3}$ and a batch size of 256. The KL loss weight is set to $1 \times 10^{-3}$.

### D.3. CIFAR-10

For the rectified flow baseline, we directly use the OT-FM and I-CFM models from (Tong et al., 2024) and evaluate their performance under different NFEs. For the consistency flow matching model, we take the public implementation from (Yang et al., 2024) and integrate the consistency loss into the same I-CFM model, naming it Consistency-FM. Additionally, we evaluate the original model from (Yang et al., 2024) with a larger parameter count, referring to it as Consistency-FM-XL.

For our V-RFM model variants, we adopt the I-CFM model from (Tong et al., 2024) and add modules to incorporate conditional signals from a 1D latent $z$. For both conditioning mechanisms discussed in Section 4.4, the sampled latent is processed through two MLP layers with SiLU activation, with both hidden and output dimensions set to 512.

In the adaptive norm variant, the latent embedding $z$ is combined with the time embedding from $v_\theta$ to regress the learnable scale and shift parameters $\gamma$ and $\beta$ for the adaptive group norm layers. For the bottleneck sum variant, the latent is added to the bottleneck feature of $v_\theta$. Since the lowest spatial resolution of the baseline network is $4 \times 4$, the 1D latent is spatially repeated and fused with the bottleneck feature via a weighted sum. To ensure effective use of the latent, we assign a weighting of 0.9 to the latent and 0.1 to the original velocity feature.

The posterior model $q_\phi$ shares a similar encoder structure to $v_\theta$ but omits the decoder. To achieve greater spatial compression, we increase the number of downsampling blocks, predicting features at a $1 \times 1$ spatial resolution. The base channel size is set to 16. Both networks are trained jointly for 600,000 iterations using the Adam optimizer with a learning rate of $2 \times 10^{-4}$ and a batch size of 128. The KL loss weighting is presented alongside the results in Table 3.

### D.4. ImageNet

We build upon the open-source SiT-XL model (Ma et al., 2024) by incorporating additional modules to integrate conditional signals from the sampled 1D latent variable $z$. The sampled latent is processed through two MLP layers with SiLU activation, with both the hidden and output dimensions set to 1152. The processed latent is then directly added to the original

conditional latent $c$, which contains timestep and class label information. The resulting conditional feature is used to predict the learnable scale and shift parameters, $\gamma$ and $\beta$, for the adaptive group normalization layers.

The posterior model $q_\phi$ shares the SiT-XL architecture but uses only half the number of transformer blocks. To achieve greater spatial compression, we apply an average pooling layer to compress the latent representation into a 1D vector, which is then processed by an MLP layer to predict $\mu_\phi$ and $\sigma_\phi$. The base channel size is set to 1152, the patch size to 2, and the number of heads to 16. Both networks are trained jointly for 800,000 iterations using the AdamW optimizer with a learning rate of $1 \times 10^{-4}$ and a global batch size of 256. The KL loss weight is set to $2 \times 10^{-3}$, and the posterior model $q_\phi$ takes $x_1$ as input. *To ensure a fair comparison, we strictly adhere to the original training recipe of SiT (Ma et al., 2024), i.e., we don't tune learning rate, decay or warm-up schedules, AdamW parameters, or employ additional data augmentation or gradient clipping during training.*

## E. Qualitative Results

### E.1. CIFAR-10

We present qualitative results of our model trained on CIFAR-10 data in Figure 15.

### E.2. ImageNet

We present qualitative results of our model trained on ImageNet data in Figure 16.

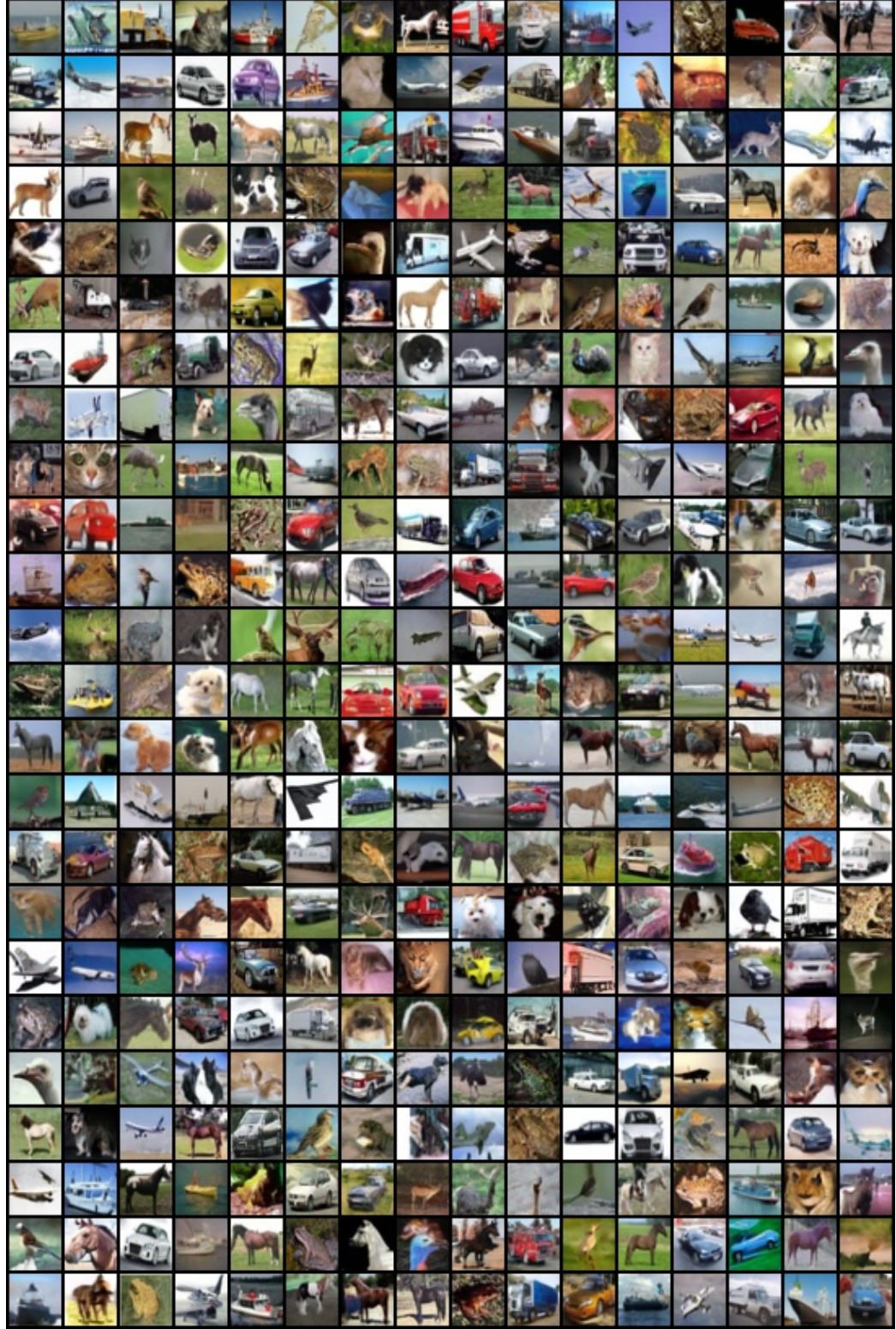

Figure 15: Randomly selected samples generated from our model trained on CIFAR-10 data.

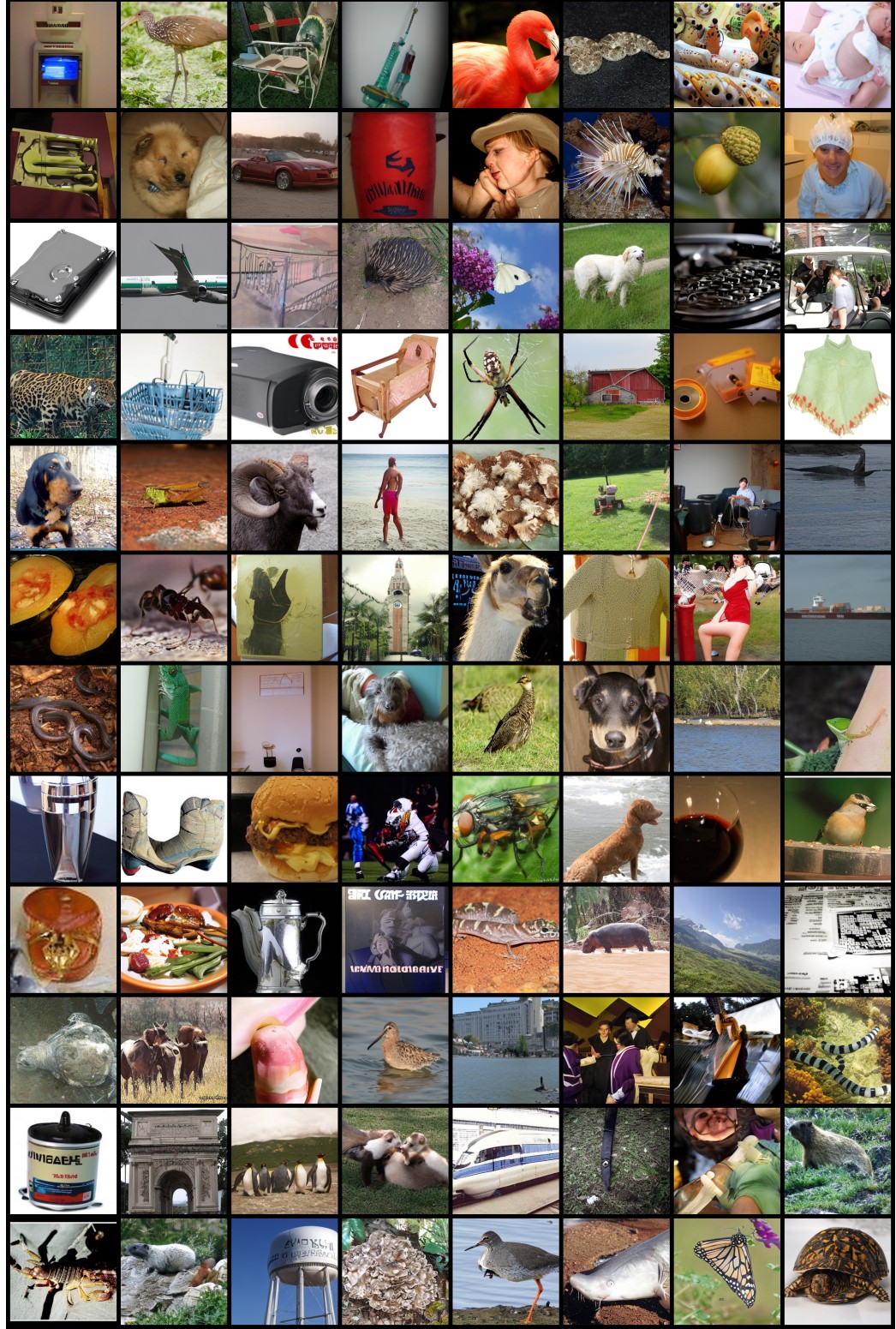

Figure 16: Randomly selected samples generated from our model trained on ImageNet data.

