# OpenReview forum: "Variational Rectified Flow Matching"
_ICML.cc/2025/Conference — ICML 2025 poster_

### Official Review · Reviewer_hji9 · 2025-02-23

**Overall Recommendation:** 4

**Summary:**

This paper introduces a variational rectified flow matching method. Instead of learning a deterministic mean velocity at time $t$, the paper explicitly models a distribution over the velocity $v_t$, grounding the approach in VAE theory.

**Claims And Evidence:**

Partially, the baselines do not include recent methods. See the detailed comments in the Weaknesses section.

**Essential References Not Discussed:**

No.

**Experimental Designs Or Analyses:**

Yes. I checked the expriments on synthetic data and CIFAR10 and ImageNet.

**Methods And Evaluation Criteria:**

Yes. They make sense.

**Other Comments Or Suggestions:**

### Suggestions:
1. In Eq. (4), $||$ should be used as separator for $q$ and $p$ in the KL divergence in stead of $|$.

### Minor questions:
1. What resources have you used for training SiT-XL on ImageNet-1k with 256x256 resolution? How long is the training schedule?

**Other Strengths And Weaknesses:**

### Strengths:

1.  The paper addresses an important problem in flow matching: the ambiguity of the marginal velocity field.

2.  The method demonstrates strong performance compared to the baseline OT-FM and I-CFM methods.

### Weaknesses:

1.  Novelty: VAEs are already well-established, which positions this work as an application of VAEs to flow matching.  It would be beneficial if the authors could further highlight the paper's unique contributions beyond the direct application of VAEs.  (To be clear, this is more of an open-ended question than a significant limitation. If VAEs are shown to be a good tool for the problem the authors address, the paper can still be considered strong.)

2.  Sampling in Algorithm 1: In Algorithm 1, how many samples of $z$ are used for large-scale experiments (e.g., ImageNet)? How does the sample size affect performance? Furthermore, during inference, are the sampled $z^{(i)}$ first averaged and then fed into the velocity predictor, or are they individually fed into the velocity predictor to predict $v^{(i)}$, which are then averaged to obtain the final velocity?  Clarifying this is crucial, as these two approaches have significantly different implications for inference speed.

3.  KL Divergence Regularization: Based on experience, VAEs are often sensitive to the choice of the KL divergence regularization weight (denoted as $\lambda$).  This sensitivity is also apparent in Table 1. The authors should provide an analysis of the impact of $\lambda$ on the learned velocity field and discuss the intuition behind its effect.

4.  Implementation Details: The authors did not provide source code, preventing a deep dive into implementation details. Specifically, there are several options for implementing the pipeline:
    (A) The encoder $q_{\phi}$ shares parameters with *a part of* the velocity predictor $v_{\theta}$.  In this case, the initial part (i.e., the first several layers) of the velocity predictor could be used to predict $\mu_{\phi}$ and $\sigma_{\phi}$, while the latter part predicts the velocity field.
    (B) The encoder $q_\phi$ uses a similar structure to $v_\theta$ but is a separate network, and they are jointly trained.

    If (B) is the case, it could introduce a substantial number of extra parameters, potentially limiting scalability. However, Table 1 shows that the total number of parameters is 37M, compared to 36.5M for the baseline.  This discrepancy needs further clarification.

5.  Related to point 4, it would be very helpful if the authors could provide a code snippet illustrating the implementation of lines 342-357.

6.  Limited Baselines: The baselines only include vanilla OT-FM and I-CFM.  Other relevant methods, such as distillation-based consistency models (Song et al. 2023) and shortcut models (Frans et al. 2024), which may also address velocity field ambiguity by encouraging "stepping over" ambiguous regions of $x_t$ with merged steps. If possible, these methods should also be considered. Including a broader range of baselines would further strengthen the paper. E.g. ``Towards Hierarchical Recitfied Flow'' from Zhang et al. (ICLR 2025) addressed a similar multimodal problem.

**Questions For Authors:**

See the questions in the Weaknesses.

**Relation To Broader Scientific Literature:**

The paper addresses the ambiguous nature of the marginal velocity field, which is critical for the performance of FM models. Previous methods tackling this challenge mainly focus on using different noise-data couplings or distillation-based methods to straighten the trajectories. This paper takes an orthogonal approach.

**Theoretical Claims:**

Yes, I checked Claim 1 and the proof.

---

> ### Author Rebuttal · Authors · 2025-04-01
>
> Thanks for detailed feedback and for recognizing the importance of addressing ambiguity in the marginal velocity field, and our strong results.
>
> **1. Paper's unique contributions**
>
> We study a method for capturing multi-modal velocity vector fields. We show that incorporating an unobserved continuous latent variable z via a variational formulation (akin to a VAE) enables the velocity model to learn a multi-modal vector field. Experiments across diverse data and models show that our method outperforms classic approaches.
>
> **2. How many z used for large-scale experiments. Averaging**
>
> To remain fair and as shown in Algorithm 2, we sample a single $z$ per data point and keep it fixed throughout integration. No averaging is performed on $z$ or predicted velocity.
>
> **3. Impact of KL weight**
>
> We summarize the impact of the KL weight $\lambda$ based on our experimental findings:
>
> 1. The model successfully captures velocity ambiguity and predicts crossing flows when $\lambda$ is in a reasonable range (in [0.1,10.0]).
> 2. When $\lambda$ is large (e.g., 100.0), the latent model is forced to equal a standard Gaussian. Hence, the latent z contains minimal useful information. Hence, the velocity network behaves similar to one obtained using classic rectified flow. This is also apparent in the loss: the KL loss diminishes, and the velocity reconstruction loss is comparable to the baseline loss. The resulting flow cannot capture ambiguity.
> 3. When $\lambda$ is small (e.g., 0.01), the model can exploit excessive information from the latent. This leads to a very low velocity reconstruction loss but a very high KL loss. The resulting flow appears as straight lines, but the endpoint distributions do not match the target data due to the mismatch between the predicted posterior and prior.
>
> In our experiments, we didn't tune the KL regularization weight much, but instead scaled the KL loss with the dimension of the latent variable. E.g., for ImageNet SiT experiments, we directly used the KL weight that we employed for CIFAR-10.
>
> **4.  Implementation of $q_\phi $and $v_\theta$**
>
> As stated in **Sec 4.4 (L342 - 343)**, $q_\phi$ and $v_\theta$ share a similar structure but are separate nets. We will clarify this to avoid confusion. Regarding the increase in parameters, as described in **Sec 3.3 and Sec 4 (L262, 348, 372 right column)**, during inference, $q_\phi$ is not used. Instead, we sample the latent variable from a prior. The only increase in parameters comes from the two MLP layers to fuse the latent z in $v_\theta$. This design ensures that our velocity network remains comparable in size to the baseline, with less than a 2% parameter increase for CIFAR-10 and less than a 0.3% increase for ImageNet.
>
> **5. Code Snippet**
>
> We will release code and models. Note, we provided more implementation details in **App D**. We are happy to address further questions during the rebuttal phase.
>
> **6. More baselines: consistency models, shortcut models, HRF**
>
> Consistency model:
> A detailed comparison to consistency models, particularly distillation models, is included in **App B**. We used the recently developed consistency flow matching [1]. It improves upon consistency models [2] and is more closely related to flow matching. We summarized the results in **App C.1**. Our key findings:
>
> * The consistency flow matching model performs well at low function evaluation regimes (i.e., with NFEs of 2 or 5).
> * Its performance degrades as NFEs increase.
> * Its best performance across all NFEs remains below classic rectified flow matching and our variational rectified flow matching.
>
> We also highlight an exciting future research direction: combining variational flow matching with consistency models, which could further enhance results.
>
> Shortcut model:
> We evaluate the Shortcut Model (XL), trained for 800k iterations, using the FID score and following the same evaluation protocol used in **Tab 2**. Our results show that our method consistently outperforms it.
>
> | | Params (M) |FID|
> | - | - | - |
> |SiT-XL| 675| 13.1|
> |Shortcut-XL| 676| 19.752 (128 NFE, reproduced)/19.630 (250 NFE, reproduced)|
> |V-SiT-XL| 677| 10.6|
> |SiT-XL (cfg=1.5)| 675| 3.43|
> |Shortcut-XL (cfg=1.5)| 676| 3.8 (128 NFE, from paper)/4.709 (128 NFE, reproduced)/4.707 (250 NFE, reproduced)|
> |V-SiT-XL (cfg=1.5) |677| 3.22|
>
> Hierarchical Rectified Flow (HRF): This concurrent work also aims to model multi-modal velocity and acceleration fields but uses a hierarchical rectified flow. Their method requires multiple integrations during inference, making it slower than our approach. Also, HRF does not support semantic disentangling of flows, as demonstrated in our **Fig 6 and 7** for MNIST and CIFAR-10.
>
> [1] Yang, L. et al. (2024) Consistency flow matching.
>
> [2] Song, Y. et al. (2023) Consistency models.
>
> **7. || should be used instead of | in KL**
>
> Thanks for spotting this typo, we'll fix.
>
> **8. resources for SiT-XL on ImageNet 256**
>
> We used 8 H100 GPUs and trained the model for about 3.5 days.

---

> > ### Comment · Reviewer_hji9 · 2025-04-06
> >
> > Thank you for the explanation. My concerns are addressed and I will raise the score. Looking forward to seeing the release of the code.

---

### Official Review · Reviewer_7qJ3 · 2025-02-26

**Overall Recommendation:** 3

**Summary:**

# Update

In the rebuttal, the authors have addressed many of my questions and criticism. I feel that the main issue has not been adequately addressed, so I decided not change the score.

To elaborate, I feel that the method adds complexity to diffusion models. The added complexity has to serve some purposes in order for it to be worthwhile. In light of the paper, the added complexity has two practical benefits.

(1) It improves the scores of over the base models.
(2) It allows a form of conditional sampling.

For (1), while improvements are consistent across sampling steps, they are not very pronounced.

* For the CIFAR-10, the scores are very close to baselines from NFE=10 onward. For NFE=2 or 5, improving from 166 to 104 or 36 to 25 cannot really be consider a practical improvement because the images produced by both the V-RFM and baselines are still of low quality. Distillation techniques, only the other hand, reduces 166 to a 1-digit figure at NFE=2 and NFE=5. This is what I meant when I said improvements cannot be compared to distillation techniques.

* For ImageNet datasets, while there is quite a significant gap when not using CFG, the gap becomes gradually smaller as training becomes longer (although the authors show that the percentage improvement still incrasese) and is significantly reduced when CFG is applied. This means that techniques already employed on non-variational models are already quite effectively, and one has to wonder whether adding a variational component to the model would worth the trouble.

Benefit (2), on the other hand, is much more interesting to me because it is a feature that a normal diffusion model has: VAE-style latent codes. This is something that I feel *worths the trouble* of making the model more complex, so I think stressing this benefit should become a bigger part of the paper. However, from reading the rebuttal, while the authors did experiments on interpolating the latents, it seems they have not investigated how to use latent codes to control the outputs with more degrees of certainly. As a result, it is unlikely that the final version of the paper would contain more material in this direction.

Because of these concerns, I decided not to change my evaluation.

# Old Summary

The paper proposes "variational rectified flow matching," an extension to (rectified) flow matching. The latter trains a neural network $v_\theta(x,t)$  that predicts the expected value of velocities induced from velocity fields that continuously transforms one Gaussian distribution (whose mean comes from a "source" distribution) to another Gaussian distribution (whose mean comes from a "target" distribution). The paper observes that there is a distribution of velocity vectors at $p(v|x,t)$ each $(x,t)$ point and explores modeling them as a part of training the flow matching model instead of just estimating the distribution's means as is done by standard rectified flow matching.

To model the velocity distribution at each point $(x,t)$, the paper casts the flow matching model as a latent variable model, much like a variational autoencoder. The flow matching model nows accept a latent variable $z$ and becomes $v_\theta(x,t,z)$. The latent is supposed to come from a prior distribution $p(z)$, taken to be the standard multivariate Gaussian distribution. To train the model, one needs to model the conditional distribution $q(z|x,t)$, which the paper models with an encoder network, muck like what VAE does.

The distribution, conditioned on the latent $z$, is modeled as a Gaussian distribution around the predicted value: $p(v|x,t,z) = \mathcal{N}(v; v_\theta(x,t,z), I)$. This gives $p(v|x,t) = \int \mathcal{N}(v; v_\theta(x,t,z), I) p(z)\, \mathrm{d}z$ where $p(z)$ is the prior distribution of $z$, which is taken to be the standard Gaussian distribution. The flow matching matching, together with the encoder, can be trained like a VAE with a loss derived from the ELBO of $\log p(v|x,t)$. The KL divergence term between $q(z|x,t)$ and $p(z)$ remains the same, but the reconstruction term in the VAE loss is replaced by the conditional flow matching loss instead. To sample with a flow matching model trained this way, one must first sample $z$ from $p(z)$, and then one can use the flow matching model to generate a sample as usual with the exception of feeding $z$ to it at every integration step.

The paper demonstrates that the method has several benefits.

(1) It yielded better evaluation scores on various datasets and models architectures compared to vanilla flow matching. In particular, the gap is wider when low number of NFEs are used to generate samples or when the training time is shorter.

(2) It yields flow matching models that can model distributions at each  $(x,t)$ point better. In aggretate, such models can model sampling trajectories, and these trajectories seem to be less curved than the non-intersecting trajectories of vanilla flow matching models.

(3) By varying $z$ at test time, one can control the generation output.

**Claims And Evidence:**

I believe claim (1) is supported by enough evidence as the paper contains experiments on 5 datasets, and three types of architecture. Performance gaps on ImageNet generation without guidance are quite significant.

Claim (2) is supported by showing that, in the 1D dataset, the vanilla flow matching model often collapes the velocity distribution. The experiment on the 2D dataset clearly shows in Figure 4(c) that the proposed method can model intersecting trajectories. However, while Figure 4(b) shows trajectories that seem to be more curved than those in Figure 4(c), it is better to quantify the average curvature of the trajectories and show the numbers along with the pictures.

For Claim (3), the paper shows that the generated samples changed when $z$ is changed in Figure 6 (MNIST dataset) and Figure 7 (CIFAR-10) datset. While one must accept that the outputs do change, it is not quite clear whether these changes are useful or intuitive. In Figure 6, different areas of the unit square seems to correspond to different digits, but the paper does not explicity show how one can obtain the desired digits through controlling $z$. In Figure 7, different latent codes seem to yield different overall brightness of the outputs, but it is unclear whether one can arbitrarily control the brightness though varying $z either. Several simple experiments where the latent codes interpolated to get the desired outcomes would make this claim stronger.

**Essential References Not Discussed:**

(1) I believe that the idea of modeling distributions of values inside a diffusion sampling process has been explored previously, and this paper is one instance of it. An example that comes to mind is the Denosing Diffusion GANs paper by Xiao et al. [1], which uses conditional GANs to model the denoising distribution at each step of the diffusion process.

(2) There is another way to combine a diffusion model with a VAE, and it involves using the former to model the latent space of the latter [2].

(3) The opposite idea to the one proposed in the paper is to regard the distribution of target values to match the neural network's output against as a kind of noise and seeks to eliminate it. Stable Target Field by Xu et al. implements this idea. [3]

1. Zhisheng Xiao, Karsten Kreis, Arash Vahdat. Tackling the Generative Learning Trilemma with Denoising Diffusion GANs. ICLR 2022.
2. Arash Vahdat, Karsten Kreis, Jan Kautz. Score-based Generative Modeling in Latent Space. NeurIPS 2021.
3. Yilun Xu, Shangyuan Tong, Tommi Jaakkola. Stable Target Field for Reduced Variance Score Estimation in Diffusion Models. ICLR 2023.

**Experimental Designs Or Analyses:**

To my understanding, the paper compares its training method, variational rectified flow matching (VRFM), against two other training methods:

(1) vanilla flow matching (OT-FM), proposed by Lipman et al. (2023), and
(2) independent coupling flow matching (I-CFM), proposed by Tong et al. (2024).

The mathematical formulation of these algorithms are slightly different, and they are almost equivalent if their source distributions are the standard Gaussian $\mathcal{N}(0,I)$. As a result, I find the comparison with I-CFM for the CIFAR-10 dataset redundant. In fact, the numbers of OT-FM and I-CFM in Table 1 are very close. Moreover, the comparison with I-CFM is only available for the CIFAR-10 dataset.

The paper would feel more consistent if either (a) comparison with I-CFM is removed or (b) it also provides comparison with I-CFM training method for other datasets.

**Methods And Evaluation Criteria:**

The method seems to make sense from the problem at hand.

The 1D and 2D datasets are used to effectively show the ability of the trained models to better capture velocities of distributions. However, I think it is better to quantify the average curvature of 2D trajectories instead of just showing pictures.

MNIST, CIFAR-10, and ImageNet are widely used datasets to benchmark generative models. The paper also uses appropriate architectures for these datasets.

The metrics (log-likelihood for 1D/2D datsets and FID for image datsets) also make sense.

**Other Comments Or Suggestions:**

I suggest replace the term "data-domain-time-domain" with "data-time space" or "$(x,t)$-space," which should be more concise.

**Other Strengths And Weaknesses:**

I believe this paper presents a new and interesting formulation of flow matching models. However, I do not feel that its benefits are compelling. Being able to model the velocity distribution is clearly a novelty, but rather a conceptual one. The most concrete one improvements in metrics which diminish as the number of function evalutions become larger. Still, for image datasets such as CIFAR-10 and MNIST (and perhaps ImageNet), these improvements are small and not comparable to improvements achieved by distillation methods.

Another benefit claimed by the paper is the ability to control the output through the latent code $z$. However, to make the paper stronger, I think the paper should do more experiments to highlight this aspect. This can include a simple method to sample a specific number from the MNIST dataset or a way to control the brightness of CIFAR-10 samples.

**Questions For Authors:**

(1) In Table 2, it seems that performance gap between SiT-XL (the baseline) and V-SiT-XL seem to diminish as training becomes longer. Can you show the FID scores at 1200K steps and/or 1600K steps to confirm that the gap still exists there as well?

(2) It would be interesting to see how ImageNet models perform at NFEs lower than 250. Please include those numberse in Table 2 if possible.

**Relation To Broader Scientific Literature:**

The paper propses a new extension to flow mathcing that allows sampling paths to be chosen based on a latent vector. In a sense, it is an interesting way to combine VAE with flow matching model.

**Theoretical Claims:**

Section 3 is easy to read and seems sound. However, the most important theoretical claim that the paper's training method preserves the marginal data distribution lacks a full formal proof. What is provided in Appendix A is a proof sketch where what is to be proven is supported by statesments such as "one can show equivalence" and "equivalence can be shown via" without any work being shown. A reader would have to go to Liu's paper and follow all the logic by themself. I suggest the authors write down the proof in Appendix A for completeness.

---

> ### Author Rebuttal · Authors · 2025-04-01
>
> Thanks for feedback and for highlighting our theoretical contributions and strong results across datasets and models.
>
> **1. Quantify the average curvature of 2D trajectories**
>
> We calculated the curvature for 2D data results (**Sec 4.2**) and find significantly lower curvature for our method:
>
> | | Mean/Max Curvature|
> | - | - |
> |Baseline (rectified flow)|	21.03/171.35|
> |Ours|	0.98/4.23|
>
> **2: How to obtain desired digits/brightness via $z$. Latent interpolation would strengthen the claim**
>
> Great suggestion. We conducted interpolation experiments and summarize findings verbally as images can't be uploaded. For MNIST, interpolating latents leads to smoothly transitioning digits (e.g., 1 → 7 → 8 → 3 → 2 → 0). Note, **Fig 6** illustrates that each digit corresponds to a specific latent $z$. Intermediate digits emerge naturally when interpolating. For CIFAR-10, we observe analogous effects—interpolating between two latents leads to smooth transitions in brightness and color patterns.
>
> **3: Marginal data distribution preservation lacks a proof**
>
> The proof in **App A** refers to results established by Liu et al. to provide adequate credit. This may require readers to reconstruct parts of the proof using work of Liu et al. To make the paper self-contained, we will expand it to add the missing steps (derivation of $d/dt E[h(X_t)]$ and equivalence of $0 = E_Z \int(...))$.
>
> **4. Formulations of OT-FM and I-CFM are almost equivalent. Redundant comparison with I-CFM**
>
> OT-FM/I-CFM differ in $x_t$ and the conditional vector field $u_t$.  Specifically, OT-FM defines $x_t$ as $\mathcal{N}(t x_1, 1-(1-\sigma) t)$ and $u_t$ is $\frac{x_1-(1-\sigma)x_t}{1-(1-\sigma)t}$. I-CFM defines $x_t$ as $\mathcal{N}(t x_1+(1-t)x_0, \sigma)$ and $u_t$ as $(x_1-x_0)$.  SiT uses the rectified flow objective, which is equivalent to I-CFM. We'll follow the suggestion to remove OT-FM for consistency.
>
> **5. Figure 12 of MNIST FID score should be table**
>
> We'll convert **Fig 12** into a table.
>
> **6. Discussions of related work**
>
> Thanks for highlighting these works. We'll add a full discussion. Below is a brief summary:
>
> 1. Denoising Diffusion GANs replace the Gaussian model in the denoising step with a multimodal distribution. Unlike our method, a conditional GAN with separate discriminator models the distribution. But GANs face mode collapse and stability issues. In contrast, our method uses rectified flow matching, preserving the maximum likelihood benefits.
> 2. Score-based Generative Modeling uses a VAE to map raw data $x_0$ into latent space $z_0$, with the VAE jointly trained with score-based generative modeling (SGM). Unlike our approach, SGM still faces ambiguity issues due to its use of a uni-modal Gaussian distribution.
> 3. Stable Target Field notes that the posterior distribution is multi-modal. To model this distribution, the paper reduces training target variance using a reference batch. In contrast, our method directly models this multi-modal posterior via a recognition model.
>
> **7. Improvements between SiT-XL and V-SiT-XL diminish as training continues**
>
> A diminishing gap is expected as absolute values decrease, a trend also presented in Fig 2 of the SiT paper (comparing the gap of DiT v.s. SiT). However, the relative improvement remains strong. We extend training to 1200k steps and report both absolute and percentage improvements. We observe the percentage improvements increase with more training iterations.
>
> ||200k	|400k	|600k	|800k	|1200k	|800k (cfg=1.5)	|1200k (cfg=1.5)|
> | - | - | - | - | - | - | - | - |
> |SiT-XL	|26.09	|17.84	|14.77	|13.15	|11.26	|3.43	|2.97|
> |V-SiT-XL	|23.34	|14.60	|12.00	|10.62	|8.97	|3.22	|2.76|
> |abs diff	|2.75	|3.24	|2.78	|2.53	|2.29	|0.21	|0.20|
> |percent diff	|10.53%	|18.16%	|18.79%	|19.24%	|20.31%	|6.12%	|6.86%|
>
> **8. Improvements are small, not comparable to distillation**
>
> We respectfully disagree. Our improvement is solid, while strictly following the open-source SiT training. Results are consistent with the SiT-to-DiT improvement, demonstrating a comparable level of progress (19.5 for DiT-XL, 17.2 for SiT-XL, and 14.6 for V-SiT-XL at 400k steps, as shown in  **Tab 2**).
>
> Also note, our key contribution is to model the velocity distribution. We find this to consistently improve evaluation metrics across all NFEs. While distillation methods may show improvements for low NFEs, our method achieves better results across both low and high NFEs.
>
> Additionally, as discussed in **App B**, V-RFM focuses on single-stage training to capture a multi-modal velocity distribution from “ground-truth” data without leveraging pre-trained models. Exploring distillation for V-RFM is an exciting avenue for future research, particularly when the interest is to improve results for low NFEs.
>
> **9. Replace "data-domain-time-domain"**
>
> Great suggestion. We'll revise.
>
>
> **10. ImageNet performance below 250 NFEs**
>
> **Fig 8** shows those results, revealing a consistent boost, further highlighting our method's effectiveness.

---

### Official Review · Reviewer_Sjgu · 2025-02-26

**Overall Recommendation:** 3

**Summary:**

The paper introduces Variational Rectified Flow Matching (VRFM), a novel approach that integrates techniques from Rectified Flow Matching (RFM) and Variational Autoencoders (VAEs). This design aims to address the vector ambiguity issue inherent in the original RFM method. Through extensive experiments, the authors demonstrate that VRFM improves data generation quality, producing samples that more closely align with ground truth compared to standard RFM. Additionally, the proposed method effectively mitigates vector ambiguity to a significant extent. Empirical evaluations on benchmark image datasets, including CIFAR-10 and ImageNet, reveal that VRFM consistently achieves superior Fréchet Inception Distance (FID) scores compared to baseline models, highlighting its effectiveness in generative modeling.

**Claims And Evidence:**

yes

**Essential References Not Discussed:**

No

**Experimental Designs Or Analyses:**

yes, there are not issues.

**Methods And Evaluation Criteria:**

yes

**Other Comments Or Suggestions:**

No

**Other Strengths And Weaknesses:**

Strengths:
	1.	The paper is well-written, with clear and easy-to-follow explanations.
	2.	The authors conduct extensive experiments to evaluate the proposed method comprehensively.
	3.	The proposed approach effectively addresses vector ambiguity and demonstrates superior generative performance compared to baseline models.

Weaknesses:
	1.	The method requires additional parameters to compute the latent representation during training, increasing computational complexity.
	2.	Some claims rely on empirical observations; providing stronger theoretical proofs would further strengthen the paper.
	3.	While the proposed method outperforms baseline models, it still lags behind the current state-of-the-art models.

**Questions For Authors:**

1.	In Figure A, the visualization depicts the ground truth data, including the source data distribution, target data distribution, and the mapping between them. I assume that the mapping between source and target data points is randomly generated, meaning any point in the source could potentially correspond to any point in the target. If this assumption is correct, then a definitive ground truth mapping may not exist. Could the authors clarify how they obtained the ground truth mapping used in the visualization?
2.	The claim that the proposed method resolves vector ambiguity is primarily supported by visualizations using toy data. Would it be possible to provide a rigorous theoretical proof to substantiate this claim?
3.	How does the choice of latent variable affect the generative path? Specifically, would the generative trajectory be different when using different numbers of latent variables during inference?

**Relation To Broader Scientific Literature:**

The proposed Variational Rectified Flow Matching (VRFM) builds upon previous work in Rectified Flow Matching (RFM) and Variational Autoencoders (VAEs). A key limitation of RFM is vector ambiguity, which can hinder generative performance. VRFM addresses this issue by integrating a variational framework, a well-established technique in generative modeling for learning more effective latent representations. Experimental results show that VRFM achieves improved Fréchet Inception Distance (FID) scores compared to RFM, indicating enhanced generative quality beyond what was previously achievable with RFM alone.

**Theoretical Claims:**

I reviewed the proofs quickly, and they appear to be correct.

---

> ### Author Rebuttal · Authors · 2025-04-01
>
> Thanks a lot for your detailed feedback and for recognizing our well-written paper, extensive experiments, and comprehensive evaluation of performance. We also appreciate the acknowledgment of V-RFM’s effectiveness in addressing velocity ambiguity and its superior performance. Below, we address questions:
>
> **1. The method requires additional parameters to compute the latent representation during training, increasing computational complexity.**
>
> Yes, extra computation is used during training. It is essential to extract latent information within the velocity network, ultimately enhancing generation quality and expressiveness. As discussed in **Sec 3.3 and 4**, the latent encoding network is not used during inference—we directly sample $z$ from the prior distribution. Furthermore, the increase in parameters is minimal (e.g., less than 0.3% on ImageNet), and the impact on inference speed is negligible while delivering superior results. The ablation study on the size of the posterior model summarized in **App Tab 5** shows that performance remains consistent across variations, demonstrating the robustness and flexibility of our approach. This allows users to balance training efficiency and runtime quality based on their computational constraints.
>
> **2. While the proposed method outperforms baseline, it lags behind the current SOTA.**
>
> The primary contribution of our work is not in surpassing the current state-of-the-art, but rather in introducing a methodological innovation, i.e., capturing the multimodal velocity field, which offers new avenues for improvement in the field. While our method lags behind the state-of-the-art, it can be combined with innovations put forth by current SOTA SiT-XL/2 + MG [1] and SiT-XL/2 + REPA [2], both built upon the SiT framework. As detailed in our experiments (**Sec 4.5, L364-367 right column**), we strictly followed the original training recipe from the open-source SiT repository and replicated the process outlined in the SiT paper for a fair comparison. Our experimental results provide empirical evidence of the effectiveness of our approach.
>
> [1] Tang, Z. et al. (2025). *Diffusion Models without Classifier-free Guidance*.
>
> [2] Yu, S. et al. (2024). *Representation alignment for generation: Training diffusion transformers is easier than you think*.
>
> **3. A definitive ground truth mapping may not exist.**
>
> A definitive “ground-truth” does not exist indeed. The (“ground-truth”) velocities form a distribution for every $(x_t, t)$-location. To see this we independently sample from both the source distribution and the target data distribution, calculate the rectified flow interpolants for each pair, and visualize them in **Fig 1** and the velocity distribution in **Fig 3(a)**. Our method models this velocity distribution at every $(x_t, t)$-location, while standard rectified flow matching cannot capture it. In the abstract and the main paper, we use quotation marks around “ground-truth” to emphasize this distinction. We will correct any missing instances in the final version.
>
> **4. Provide a rigorous theoretical proof to substantiate the claim that the proposed method resolves vector ambiguity.**
>
> In **Sec 3**, we show that the proposed approach leads to a mixture model for the velocity distribution (**L179-180**). A mixture model is theoretically capable of capturing multi-modality. Following classic expectation maximization or variational inference, we introduce the recognition model, derive the lower bound of the marginal likelihood for an individual data point (**L189-192**), and present the variational flow matching objective (**L197-199**). To further substantiate our approach, in **App A**, we show how to prove that the distribution learned by the variational objective preserves the marginal data distribution. Empirically, on 1D data we visualize the learned velocity distribution in **Fig 3**, showing that the method indeed learns the velocity ambiguity. Further, during training on high-dimensional data, we observe that our method achieves better velocity reconstruction losses (**App Fig 15**) compared to standard rectified flow, indicating that the predicted velocities more accurately approximate the “ground-truth” velocities.
>
> **5. How does the choice of latent variable affect the generative path.**
>
> We studied the role of $z$ for MNIST data (**Fig 6**) and CIFAR-10 data (**Fig 7**). As noted in **Sec 4.3 and 4.4**, we observe clear patterns in the generated samples based on $z$. Specifically, images conditioned on the same latent $z$ exhibit consistent color patterns, while images at the same grid location show similar content. These observations validate the effectiveness of the latent variable $z$ in influencing and controlling the generated samples.

---

### Official Review · Reviewer_8Z7u · 2025-03-14

**Overall Recommendation:** 3

**Summary:**

This paper proposes Variational Rectified Flow Matching (V-RFM), a generative model that integrates Variational Autoencoders (VAEs) with Rectified Flow Matching (RFM). Unlike conventional RFM, which struggles to capture the multimodal nature of the ground-truth velocity vector field and learns only a single averaged direction, V-RFM introduces an encoder-based architecture to enable modeling of multimodal velocity fields. This capability allows V-RFM to theoretically achieve straighter sampling trajectories compared to traditional RFM frameworks.

## update after rebuttal
I thank the authors for their response and I will maintain my score as Weak Accept. I suggest that the authors add discussion and evidence on training stability and convergence in an updated paper. Also, it would be valuable to report more results in an updated version.

**Claims And Evidence:**

Yes, the major claim of this submission is supported by experiments.

**Essential References Not Discussed:**

The paper provides a full discussion of related work.

**Experimental Designs Or Analyses:**

I suggest adding more baselines in the experiments in Tables 1 and 2, such as flow matching and 1/2/3-Rectified Flow, which can increase the soundness/validity of experimental designs. It is also recommended to increase qualitative comparison with those baselines.

**Methods And Evaluation Criteria:**

Yes, the proposed method makes sense for the current problem.

**Other Comments Or Suggestions:**

N/A

**Other Strengths And Weaknesses:**

Strengths
+ The proposed V-RFM is novel for me, and is promising as a generative model.
+ The introduction of encoder enables V-RFM to have the ability to infer latent codes given the data samples.

Weaknesses
- I am more concerned about the training stability and effectiveness of the method as a generative model. V-RFM introduces VAE to RFM. Relatively speaking, the training stability and effectiveness of VAE are not as good as RFM because it may encounter the posterior collapse problem. However, this paper does not seem to discuss whether the encoder will encounter this problem and how it affects the generation effect.

**Questions For Authors:**

Please see Weaknesses and Experimental Designs Or Analyses.

**Relation To Broader Scientific Literature:**

No.

**Theoretical Claims:**

I checked the correctness of the proof of Claim 1 of the main paper. The proof looks correct to me.

---

> ### Author Rebuttal · Authors · 2025-04-01
>
> Thanks a lot for your detailed feedback and for recognizing V-RFM's novelty and promise as a generative model. See below for answers to your questions:
>
>
> **1. Training stability of V-RFM. VAE training is not as stable as RFM because it may encounter the posterior collapse problem.**
>
> Great question. During our studies we observed that stability can be controlled by the architecture. Specifically, as stated in **L353-357**, bottleneck sum, which fuses the latent $z$ with activations of the velocity network, and adaptive normalization, which explicitly scales and offsets the latent $z$ at multiple layers of the velocity network, are effective at ensuring that the latent variable $z$ sampled from the posterior is not ignored. As shown in **App Fig 15**, the reconstruction losses of V-RFM remain consistently lower than the baseline, demonstrating that the latent variable contributes meaningfully to reducing the reconstruction loss. Furthermore, **Fig 6 and 7** confirm that modifying the latent $z$ alters the predicted image, further verifying that our latent representation remains informative and utilized throughout training. Lastly, we conducted an ablation study on the size of the posterior model, summarized in **App Tab 5**. The results show that performance remains consistent across variations, indicating that even with a very small encoder (6.7% of its original size), the latent information remains informative and helps achieve competitive FID scores.
>
>
> **2. More baselines in Tables 1 and 2, such as flow matching and 1/2/3-Rectified Flow.**
>
> Note, the OT-FM/I-CFM baselines in **Tab 1** and the SiT baseline in **Tab 2** employ the flow matching objective with differences in the parameterization of $x_t$ and the conditional vector field $u_t$.  Specifically, OT-FM parameterizes $x_t$ as $\mathcal{N}(t x_1, 1-(1-\sigma) t)$, while I-CFM defines it as $\mathcal{N}(t x_1+(1-t)x_0, \sigma)$. The conditional vector field $u_t$ is $\frac{x_1-(1-\sigma)x_t}{1-(1-\sigma)t}$ for OT-FM, while it is simply $(x_1-x_0)$ for I-CFM.  We also note that 1-Rectified Flow is equivalent to I-CFM.
>
> We have added a comparison with 2/3-Rectified Flow via Reflow. We find that while strong FID scores in the low-NFE regime are achieved, it does so at the cost of limiting peak performance at high NFE. We emphasize that Reflow is a supplementary technique applied on top of Rectified Flow Matching (RFM)—primarily aimed at fast sampling rather than improved sample quality. It also requires $N$ times longer training and a significantly larger fine-tuning dataset, where $N$ denotes the number of Reflow rounds. These differences make a direct comparison with our V-RFM less fair. Hence, RFM without Reflow is a more appropriate baseline. Additionally, Reflow can be applied to our method as well, potentially improving results at the cost of increased training overhead. We will clarify these points in the final version and include additional qualitative comparisons for a more precise evaluation.
>
> |Methods    |# Params    |2    |5    |10    |50    |100    |1000    |Adaptive|
> |:---------:|:--------:|:--------:|:-----------:|:-----------:|:-----------:|:-----------:|:-----------:|:-----------:|
> |RFM w.o. Reflow / I-CFM    |36.5M    |168.654    |35.489    |13.788    |5.288    |4.461    |3.643    |3.659|
> |RFM w. 1 Reflow            |36.5M    |7.512    |5.906    |5.513    |5.283    |5.276    |5.276    |5.275|
> |RFM w. 2 Reflow            |36.5M    |7.559    |6.925    |6.776    |6.729    |6.733    |6.752    |6.752|

---

### Decision · Program_Chairs · 2025-05-01

**Decision:**

Accept (poster)

**Comment:**

After the rebuttal and discussion phases, the paper received scores of 4, 3, 3, and 3, which exceed the expected threshold for acceptance. After briefly reviewing the comments and the authors' responses, I believe the paper meets the acceptance criteria for ICML.

However, I strongly encourage the authors to address the unresolved concerns raised by the reviewers—such as the theoretical explanations pointed out by Reviewer Sjgu, the remaining issues raised by Reviewer 7qJ3, and the request for code release made by Reviewer hji9. These revisions would help strengthen the paper and ensure its contributions are more clearly presented.